# A pan-immunotherapy signature to predict intratumoral CD8$^+$ T cell expansions

Munetomo Takahashi ®[1,2,6] ✉, Mikiya Tsunoda ®[3,6], Hiroyasu Aoki[2,3,4], Masaki Kurosu[3], Haru Ogiwara[3], Shigeyuki Shichino ®[3], David Bending ®[5], Shumpei Ishikawa[2], James E. D. Thaventhiran ®[1], Kouji Matsushima ®[3] & Satoshi Ueha ®[3] ✉

Effective cancer immunotherapy relies on the clonal proliferation and expansion of CD8$^+$ T cells in the tumor. However, our insights into clonal expansions are limited, owing to an inability to track the same clones in tumors over time. Here, we develop a multi-site tumor mouse model system to track hundreds of expanding and contracting CD8$^+$ T cell clones over multiple timepoints in tumors of the same individual. Through coupling of clonal expansion dynamics and single-cell RNA/TCR-seq data, we identify a transcriptomic signature in PD-1$^+$Ly108$^+$ precursor exhausted cells that strongly predicts rates of intratumoral clone expansion. The signature correlates with expansion in mice, both with and without immunotherapies, and in patients undergoing PD-1 blockade therapy. Expression of the signature during treatment corresponds with positive clinical outcomes. Downregulation of the signature precedes clone contraction—a phase in which clones contract but maintain revivable precursor exhausted cells in the tumor. LAG-3 blockade reactivates the expansion signature, re-expanding pre-existing clones, including previously contracted clones. These findings reveal how the study of clonal expansion dynamics provide a powerful 'pan-immunotherapy' signature for monitoring immunotherapies with implications for their future development.

Immunotherapies, centered around immune checkpoint blockade (ICB) therapies, have revolutionized cancer treatment by harnessing the power of our immune response[1]. Despite their success, challenges remain, particularly in overcoming treatment-resistant cancers[2]. While immunotherapies, including CTLA-4, PD-1/PD-L1 and LAG-3 blockade, demonstrate clinical potential[3–6], their development is hindered by our inability to comprehensively monitor their effect on T cell responses over time[2,4,7,8], complicating efforts to accurately predict patient outcomes and optimize treatment protocols.

Recent landmark papers sampling tumors pre and post PD-1/PD-L1 blockade show that PD-1/PD-L1 blockade recruits novel T cell clones and revives pre-existing clones at the tumor site[9–12], highlighting the need to track T cell responses in the same individuals to understand the effects of immunotherapy[2,7,8]. These studies have relied on longitudinally sampling the same patient tumors, limiting their broader application for investigating emerging immunotherapies, for which such patient samples are sparse[13]. Additionally, these studies only sample a small portion of the tumor, making it

[1]Medical Research Council Toxicology Unit, University of Cambridge, Gleeson Building, Tennis Court Road, Cambridge CB2 1QR, UK. [2]Department of Preventive Medicine, Graduate School of Medicine, The University of Tokyo, Tokyo, Japan. [3]Division of Molecular Regulation of Inflammatory and Immune Diseases, Research Institute for Biomedical Sciences, Tokyo University of Science, Chiba, Japan. [4]Department of Host-Microbe Interactions, St. Jude Children's Research Hospital, Memphis, TN, USA. [5]Department of Immunology and Immunotherapy, School of Infection, Inflammation and Immunology, College of Medicine and Health, University of Birmingham, Birmingham B15 2TT, UK. [6]These authors contributed equally: Munetomo Takahashi, Mikiya Tsunoda. ✉e-mail: munetomo.takahashi@gmail.com; ueha@rs.tus.ac.jp

difficult to determine whether any observed changes reflect true biology, or are an artifact of the limited sampling[12]. While preclinical models have the potential to fill this gap−as demonstrated by a recent study time-stamping tumor-infiltrating CD8+ T cells[13], they are also currently limited by their inability to comprehensively sample tumors over time.

Here, we develop a multi-site tumor mouse model system to track CD8+ T cell clones in tumors of the same individual over time. We inoculate mice with multiple tumors to overcome the challenges of sampling cells from the same tissue and utilized the T-cell receptor (TCR) sequence as a natural genetic barcode to identify clonally related T cell populations ("clones") across tumors. By sequentially sampling tumors from the same mice, we successfully track hundreds of expanding and contracting clones over multiple timepoints in tumors of the same individual. We identify a transcriptomic CD8+ T cell signature that is highly predictive of clone expansion. This expansion signature successfully predicts rates of intratumoral clone expansion in mice, both with and without immunotherapies, and in patients undergoing PD-1 blockade therapy. Expression of the signature during treatment correlates with favorable clinical outcomes in these patients. We observe that LAG-3 blockade upregulates the expansion signature, re-expanding pre-existing clones, including contracted clones. Together, these findings provide a time-resolved understanding of CD8+ T cell clonal responses that provide a powerful 'pan-immunotherapy' signature for monitoring immunotherapies, with implications for their future development.

## Results

### A time-resolved model of CD8+ T cell responses tracks expanding and contracting clones in the tumor

We, and others, have previously demonstrated that tumor growth and T cell infiltration is mirrored across bilaterally inoculated tumors[14–17]. We therefore investigated if a multi-site tumor mouse model could be utilized to track T cell responses over time in the same individual. Lewis lung carcinoma (LLC) cells were inoculated into the left flank, right flank, and left hip of mice. The characteristics of tumor-infiltrating CD8+ T cell, as evaluated by PD-1, Ly108 and TIM-3 surface markers, were largely unaffected by the number of tumors implanted (Supplementary Fig. 1a, b). Across all tumors, including the left and right flank tumors, the sizes of any given clone (as identified by the read frequency of a TCR sequence in the repertoire) were highly correlated (Fig. 1a-b, Supplementary Fig. 1c). These results indicate that for a given clone, each tumor acts as a proxy of clone size in the other tumors. We reasoned that comparison of clones across sequentially removed tumors could therefore enable the tracking of clones over time (Fig. 1c). LLC cells were bilaterally inoculated into the left and right flanks of mice (*n* = 7). After 14 days, the left tumor was surgically removed, and a week later, the right tumor was harvested with the mice. CD8+ T cells obtained from the left and right tumor were analyzed by bulk TCR-sequencing (TCR-seq) to identify clones. Compared to clones obtained from tumors removed at the same time, clones obtained from sequentially removed tumors had greater variance in size, indicating ongoing expansion and contraction (Fig. 1d). Following established protocols[18–20], we applied a beta-binomial model to define significantly expanded and contracted clones. Clones that were significantly larger in the tumor harvested at the later timepoint (right tumor) were defined "expanding clones" while clones that were significantly smaller were defined "contracting clones" (Fig. 1d, Supplementary Fig. 1d). Expanding and contracting clones were reflected by changes in cell count and identified in all mice (Fig. 1e, f, Supplementary Fig. 1e, f), indicating our system could reproducibly track clonal expansion dynamics.

### Precursor exhausted cells drive clone expansion and are preferentially maintained in the tumor during clone contraction

To characterize the differentiation states of expanding and contracting clones, we sorted intratumoral CD8+ T cells based on PD-1, Ly108 and TIM-3 expression before processing them through bulk TCR-seq (Fig. 2a, Supplementary Data 1). This analysis identified the proportion of cells in each clone that were PD-1−, precursor exhausted (PD-1+Ly108+TIM-3−)−a stem cell like state capable of self-renewal[21,22], intermediate exhausted (PD-1+Ly108+TIM-3+) and terminally exhausted (PD-1+Ly108−TIM-3+)−a short-lived terminally differentiated state[21] on days 14 and 21. Following accumulating evidence that terminally exhausted cells in tumors are specifically derived from tumor-specific T cells[10,23,24], we followed published protocols and filtered for clones that contained counts in the terminally exhausted compartment to enrich for tumor reactive clones[10] (Supplementary Fig. 2a). First, to assess whether sequential tumor removal influenced the clonal dynamics of tumor-specific T cells in the remaining tumor through antigen spreading, we blocked lymphocyte migration using FTY720[25] following resection of the first tumor. FTY720 blockade had no effect on the fraction or frequency of expanding and contracting clones (Supplementary Fig. 2b).

We next compared the differentiation states of expanding and contracting clones. On day 14, expanding clones contained a higher fraction of precursor exhausted cells, and a lower fraction of terminally exhausted cells than contracting clones (Supplementary Fig. 2b, c). Over time, expanding clones decreased their fraction of precursor exhausted cells and increased their fraction of terminally exhausted cells (Fig. 2b), consistent with the differentiation of precursor exhausted cells into terminally exhausted cells[21]. In contrast, contracting clones, particularly those with high fractions of terminally exhausted cells 'regressed'−they decreased their fraction of terminally exhausted cells and increased their fraction of precursor exhausted cells. We analyzed changes in the number of precursor exhausted and terminally exhausted cells in each clone (Fig. 2c). For many clones with high fractions of terminally exhausted cells, the number of terminally exhausted cells decreased while the number of precursor exhausted cells decreased less, or in some cases, increased (Fig. 2d). Collectively, these results indicate that precursor exhausted cells differentiate to drive clone expansion and are preferentially maintained in the tumor during clone contraction (Fig. 2e).

### A transcriptomic signature predicts intratumoral clone expansion

Since both expanding and contracting clones contained similar numbers of precursor exhausted cells per clone (Supplementary Fig. 2d), we hypothesized that differences in their expansion dynamics could be attributed to the state of their precursor exhausted cells. We utilized the same bilateral tumor excision schema and performed single-cell RNA/TCR-seq on a fraction of T cells, enriched for precursor exhausted cells, in addition to bulk TCR-seq (Supplementary Fig. 3a, Supplementary Data 1). This produced a time-resolved dataset that annotated single CD8+ T cells with their transcriptomic states and their clonal expansion dynamics.

Unsupervised clustering of the single-cell RNA/TCR-seq data retrieved precursor exhausted (clusters 0, 1 and 4), intermediate exhausted (clusters 2, 5, 7 and 8), terminally exhausted (cluster 3) and proliferating cells (cluster 6−of which the majority were terminally exhausted cells) (Fig. 3a, Supplementary Fig. 3b-c), which we confirmed by projection on a reference atlas of tumor-infiltrating T cells[26] (Supplementary Fig. 3d) and by pseudo-time analysis (Supplementary Fig. 3e). We compared the transcriptomic state of precursor exhausted cells from expanding and contracting clones on day 14 to find markers that would predict their future clonal expansion dynamics (Fig. 3b). Precursor exhausted cells from expanding and contracting clones

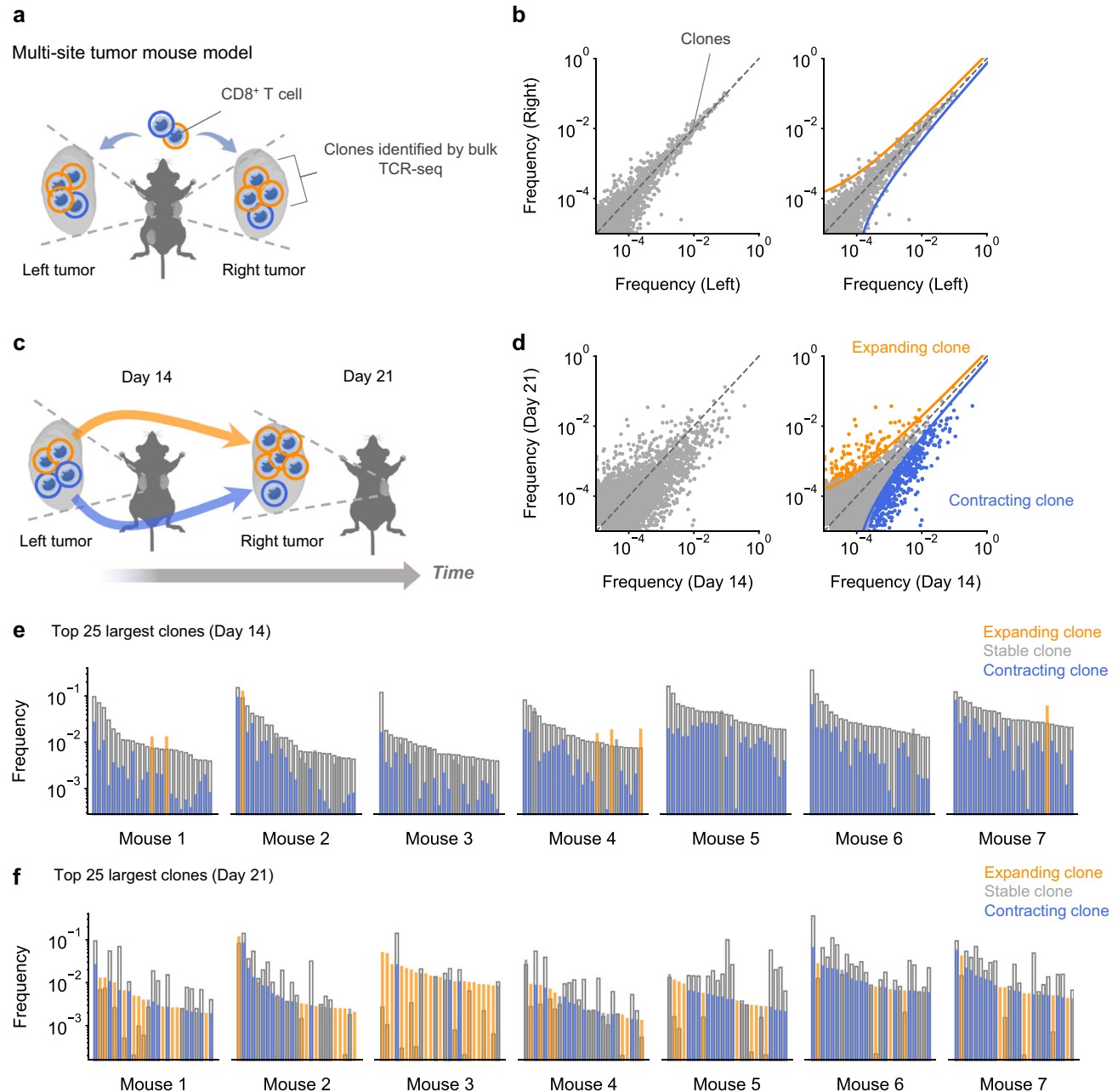

**Fig. 1 | A time-resolved model of CD8+ T cell responses tracks expanding and contracting clones in the tumor. a** Depiction of a multi-site tumor mouse model: Lewis lung carcinoma tumors are implanted simultaneously into the left flank, right flank and left hip of mice. Cells from the tumors are analyzed by bulk TCR-sequencing to identify cells belonging to the same clone. **b** Scatter plots displaying the frequency of clones (normalized read count of each clone) in the left and right flank tumors of 3 mice excised at the same time (Left) with representative bounds defining expanding and contracting clones overlayed (Right). **c** Bilateral tumors were excised independently on sequential days in the same mice to track the temporal dynamics of clones in the tumor. **d** Scatter plots displaying the frequency of clones in tumors on days 14 and 21 from 7 mice (Left) with representative bounds defining expanding and contracting clones overlayed (Right). Clones colored by their expansion dynamics as defined using a beta-binomial model from Rytlewski et al.[18]. **e, f** The top 25 largest clones obtained from each mouse on day 14 (**e**) and 21 (**f**) with the frequency of each clone on day 14 (empty boxes) and 21 (boxes colored by the clone's expansion dynamics). Dots represent clones (**b, d**). Illustrations created with cartoons from BioRender (Ueha, S. (2025) https://BioRender.com/yx373li) and Irasutoya.com (**a, c**). Source data are provided as a Source Data file.

were mainly enriched in a precursor exhausted cluster (cluster 0) that highly expressed *Lag3* and lowly expressed *Sell* (Fig. 3c, Supplementary Fig. 3c). These cells could not be differentiated based on their Uniform Manifold Approximation and Projection (UMAP) representation (Fig. 3c, Supplementary Fig. 3f). We first performed single cell weighted gene network correlation analysis (scWGCNA)[27,28]—a method that clusters modules of gene networks based on co-expression similarity for an unbiased assessment of the biological states hidden within this precursor exhausted cluster. scWGCNA revealed 2 modules

(Module 1 and 2). Module 1 contained genes associated with calcium ion release (Ryr2, Trdn) and DNA repair (Aptx, Ankhd1). Module 2 predominantly contained ribosomal genes, and gene ontology (GO) assessment associated these genes strongly with translation (Supplementary Data 2). We scored cells from expanding and contracting clones for expression of genes from each module. Intriguingly, Module 1 was highly expressed in cells from contracting clones whereas Module 2 was highly expressed in cells from expansion clones

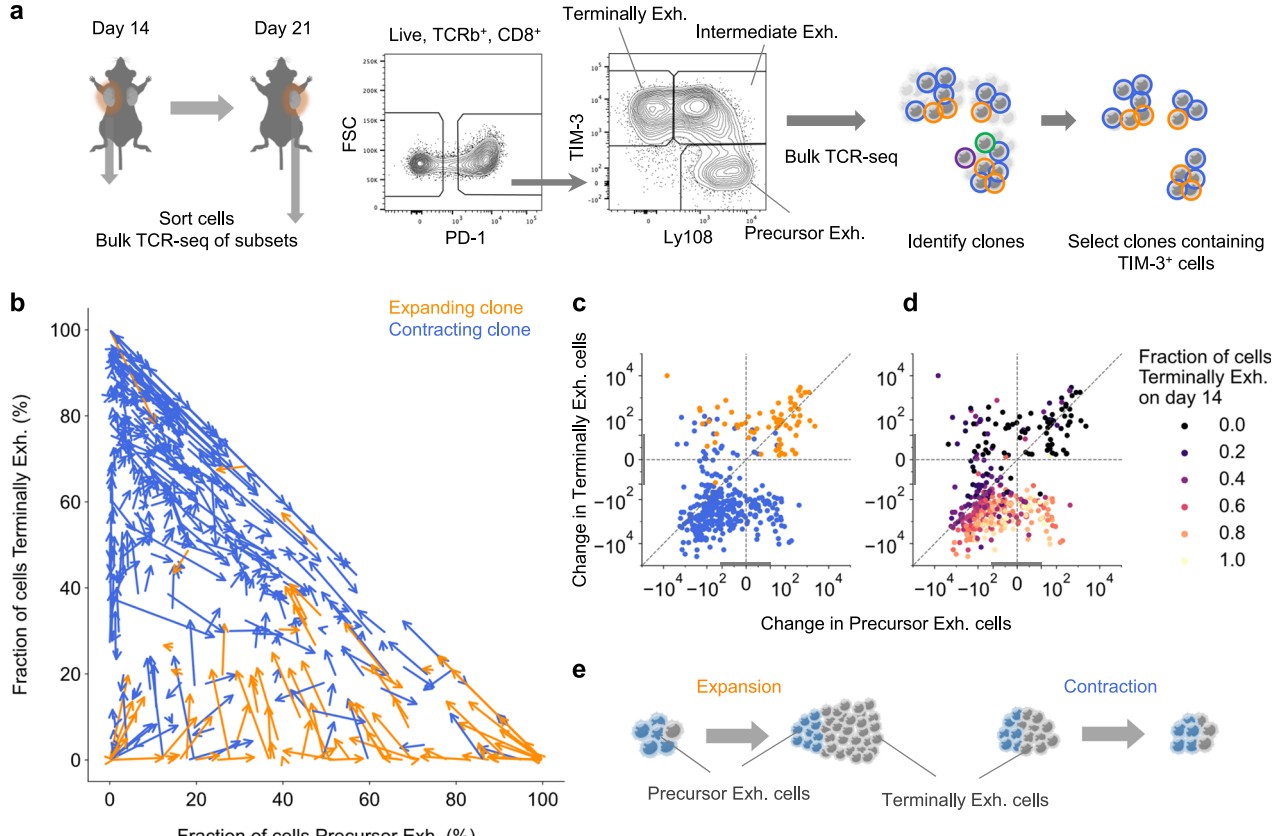

**Fig. 2 | Precursor exhausted cells drive clone expansion and are preferentially maintained in the tumor during clone contraction. a** Cells from tumors excised on days 14 and 21 ($n = 7$ mice) were sorted based on marker expression before processing through bulk TCR-sequencing, enabling phenotypic characterization of individual clones. Clones containing TIM-3$^+$ cells were selected for downstream analysis as in Supplementary Fig. 2a. **b** Velocity plot depicting the cellular composition changes of expanding and contracting clones from day 14 to 21. Arrows depict individual clones. The base of the arrow indicates the fraction of cells that were precursor exhausted and terminally exhausted on day 14 while the arrow direction and length indicate the magnitude and vector of change in these fractions over time. **c**, **d** Change in the number of precursor and terminally exhausted cells between day 14 and 21 colored by their expansion dynamics (**c**) and fraction of cells terminally exhausted on day 14 (**d**). Symmetrical log axis with a linear range of -20 to +20 (as depicted by gray lines on axes). **e** Expansion and contraction phases of clones. Expanding and contracting clones from the 7 mice that displayed changes in both the precursor exhausted and terminally exhausted cell fractions are shown (**b**–**d**). Dots represent clones (**c**, **d**). Illustrations created with cartoons from BioRender (Ueha, S. (2025) https://BioRender.com/yx373li) and Irasutoya.com (**a**). Source data are provided as a Source Data file.

(Supplementary Fig. 3g). These results suggested that Tpex cells from expanding and contracting clones were in distinct states.

Differential gene expression (DEG) analysis between precursor exhausted cells from expanding and contracting clones revealed that cells from expanding clones overexpressed 45 genes of which 24 intersected with Module 2 genes (Supplementary Fig. 4a). Out of the 45 genes, 22 were canonical genes (and not mitochondrial or ribosomal genes) which we termed the 'expansion' signature (Fig. 3d, Supplementary Tables 2 and 3). The expansion signature contained genes associated with T cell memory states (*Tcf7, Ccr7*), T cell signaling (*Ptprcap, Cd160*), and cell cycle regulation (*Hmgn1, Ddit4, Rack1*). As expected, precursor exhausted cells from clones that later expanded upregulated the expansion signature while cells from clones that later contracted downregulated the signature (Supplementary Fig. 4b). We tested if, in addition to predicting expansion or contraction of the whole clone, expression of the expansion signature could also predict the extent of clone expansion (the log2 fold-increase in clone frequency from day 14 to 21). Strikingly, the expansion signature score correlated with clone expansion (Fig. 3e). The correlation was stronger when we scored the expansion of individual clones by averaging the score of their constituent precursor exhausted cells (Fig. 3f). The correlations were evident when we analyzed each mouse independently (Supplementary Fig. 4c) and altered the analysis criteria

(Supplementary Fig. 4d). Other previously reported precursor exhausted cell state gene signatures from mice and humans[29–32], including from a recently described CD62L$^+$ precursor exhausted population[32], did not consistently correlate with the expansion of clones (Supplementary Fig. 4e, f). Lastly, while the expansion signature was expressed most highly in the precursor exhausted state (Supplementary Fig. 4g), expression of the signature in in both intermediate exhausted cells and terminally exhausted/proliferating cells correlated strongly with clonal expansion (Supplementary Fig. 4h).

To undertake an unbiased assessment of the pathways associated with the expansion signature, we scored precursor exhausted cells of clones for hallmark pathway gene signatures[33]—50 gene signatures of well-defined biological processes, and searched for correlations with the expansion signature score at the clonal level. Hallmark pathways associated with increased metabolism (oxidative phosphorylation, glycolysis, adipogenesis, fatty acid metabolism) and proliferation (MYC targets, MTORC1 signaling) positively correlated with the expansion signature score, whereas a pathway for TGF-β signaling inversely correlated with the expansion signature score (Supplementary Fig. 5a). Recent studies have implicated a role for TGF-β and Let-7 signaling in repressing cellular metabolism through inhibition of MYC signaling[34–36]. In our dataset, clones with low expansion signature scores highly expressed genes associated with TGF-β signaling in

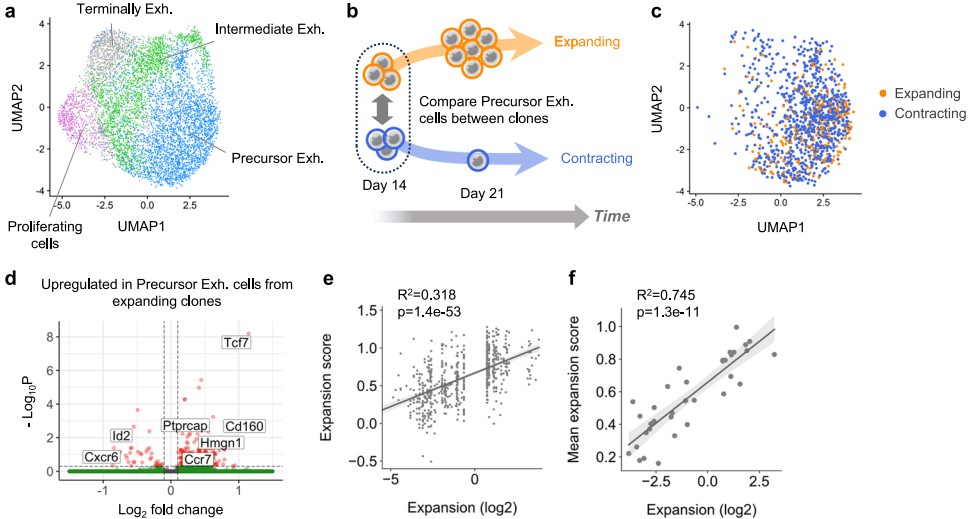

**Fig. 3 | A transcriptomic signature predicts intratumoral clone expansion.** Cells from bilateral tumors excised on days 14 and 21 from 3 mice were processed by a combination of single-cell RNA/TCR-sequencing and bulk TCR-sequencing to enable integration of single-cell transcriptomic data of clones with their expansion dynamics. **a** Uniform Manifold Approximation and Projection (UMAP) map of 2928 cells from tumors excised on day 14, colored and labeled by cell type. **b**–**f** Precursor exhausted cells from tumors excised on day 14, belonging to expanding and contracting clones, were investigated for transcriptomic signatures that could predict future clone expansion. **c** UMAP map of precursor exhausted cells colored by their expansion dynamics. **d** Differential gene expression analysis of genes overexpressed in precursor exhausted cells from expanding clones over contracting clones. Notable genes are labeled. **e** Scatter plot comparing the expansion signature score with the expansion (the log2 fold-increase in clone frequency from day 14 to 21) of the clone for individual cells. Cells belonging to the same clone have the same expansion rate. **f** Scatter plot comparing the mean expansion signature score with the expansion of clones. Clones were scored for their expression of the expansion signature in constituent precursor exhausted cells on day 14. Analysis on clones with at least 5 precursor exhausted cells on day 14. Dots represent genes (**d**), cells (**a**, **c**, **e**) and clones (**f**). Scatterplots are displayed with a line indicating the linear regression model fit and 95% confidence intervals obtained by bootstrapping (**e**, **f**). Statistical testing via two-sided Wilcoxon rank-sum test corrected with the Benjamini–Hochberg procedure (**d**) and two-sided Wald test (**e**, **f**) (****$p < 0.0001$; ***$p < 0.001$; **$p < 0.01$; *$p < 0.05$; ns $p > 0.05$). Illustrations created with cartoons from Iraustoya.com (**b**). Source data are provided as a Source Data file.

T cells[35,37] whereas clones with high expansion signature scores highly expressed genes inhibited by Let-7[36] (Supplementary Fig. 5c). These findings were reflected in precursor exhausted cells from a single-cell RNA/TCR-seq dataset of antigen-signaled CD8+ T cells infiltrating the YUMMER1.7 model[38] (Supplementary Fig. 5b, d). Taken together, these results demonstrate the identification of an expansion signature, associated with high metabolic activity and reduced TGF-β signaling, that is highly predictive of the extent of future clone expansions.

### The expansion signature predicts intratumoral clone expansion during immunotherapy

To establish the utility of the expansion signature, especially in the context of immunotherapy, we tested whether clonal expansion could be predicted from the expansion signature in both untreated and immunotherapy treated conditions (Fig. 4a). Using the multi-site tumor model, we generated additional time-resolved scRNA/TCR-seq datasets to track clonal responses under anti-PDL1/CTLA4 and anti-LAG-3 blockade. The new datasets enabled us to match the transcriptomic states of single CD8+ T cells before therapy, with their clonal expansion dynamics after therapy (Supplementary Fig. 6a). As before, unsupervised clustering of the single-cell RNA/TCR-seq data retrieved precursor exhausted, intermediate exhausted, terminally exhausted and proliferating cells (of which the majority were terminally exhausted cells) (Supplementary Fig. 6b-f). To approximate how clonal expression profiles would appear in non-enriched or human datasets, we calculated each clone's overall expansion signature score as the weighted average of its state-specific scores, using the proportion of clonal cells in each state on day 14 as weights. Strikingly, in untreated and in both anti-PDL-1/CTLA-4 and LAG-3 conditions, the clones' expansion signature score on day 14 correlated strongly with their future expansion (Fig. 4b), indicating that the expansion signature could be used to predict clone expansion during immunotherapy.

We explored whether the expansion signature might be applicable to humans. We analyzed single-cell RNA/TCR-seq datasets obtained from site-matched longitudinal biopsies of basal cell carcinoma (BCC)[9], head and neck squamous cell carcinoma (HNSCC)[20], breast cancer (BC)[12] and non-small cell lung cancer (NSCLC)[10] patient datasets pre and post anti-PD-1 therapy (Fig. 4c). In all the datasets, clones that expanded over time had higher expression of the expansion signature pre-therapy (Fig. 4d). Comparing responders and non-responders, clones from responders in post-treatment samples had significantly higher expansion signature scores than clones from non-responders (Fig. 4e). No such difference was observed in pre-treatment samples where comparisons between responders and non-responders were possible (Supplementary Fig. 7a). These results suggested that response is associated with induced clonal expansion activity, leading us to query whether the expansion signature could be used to monitor immunotherapy outcomes.

We scored bulk RNA-seq melanoma tumor biopsies of pre- and on- PD-1/PD-L1 blockade treatment[39] with the expansion signature score. We utilized a stringent expansion gene signature (adjusted p-value cut-off of 0.01 instead of 0.05, Supplementary Data 2) to account for the increased biological variability in tumor biopsies. Corroborating our previous analysis, on-treatment (but not pre-treatment) biopsies from responders scored significantly higher for the expansion signature than non-responders (Supplementary Fig. 7b). Patients in this dataset were either treatment naive or were treated with CTLA-4 blockade prior to PD-1/PD-L1 blockade treatment. Elliot et al.[40] recently reported a TCR.strong metric (consisting of *TNFRSF4*, *IRF8*, *STAT4*, *TNIP3* and *ICOS* derived from strong TCR signaling) that could stratify patient outcomes for treatment naïve patients in this dataset. Following their protocols[40], we split patients into a "High" or "Low" cohort based on the median expansion signature score and (for comparison) TCR.strong score and plotted corresponding survival curves (Fig. 4f). Strikingly, the "High" expansion signature score cohort showed a

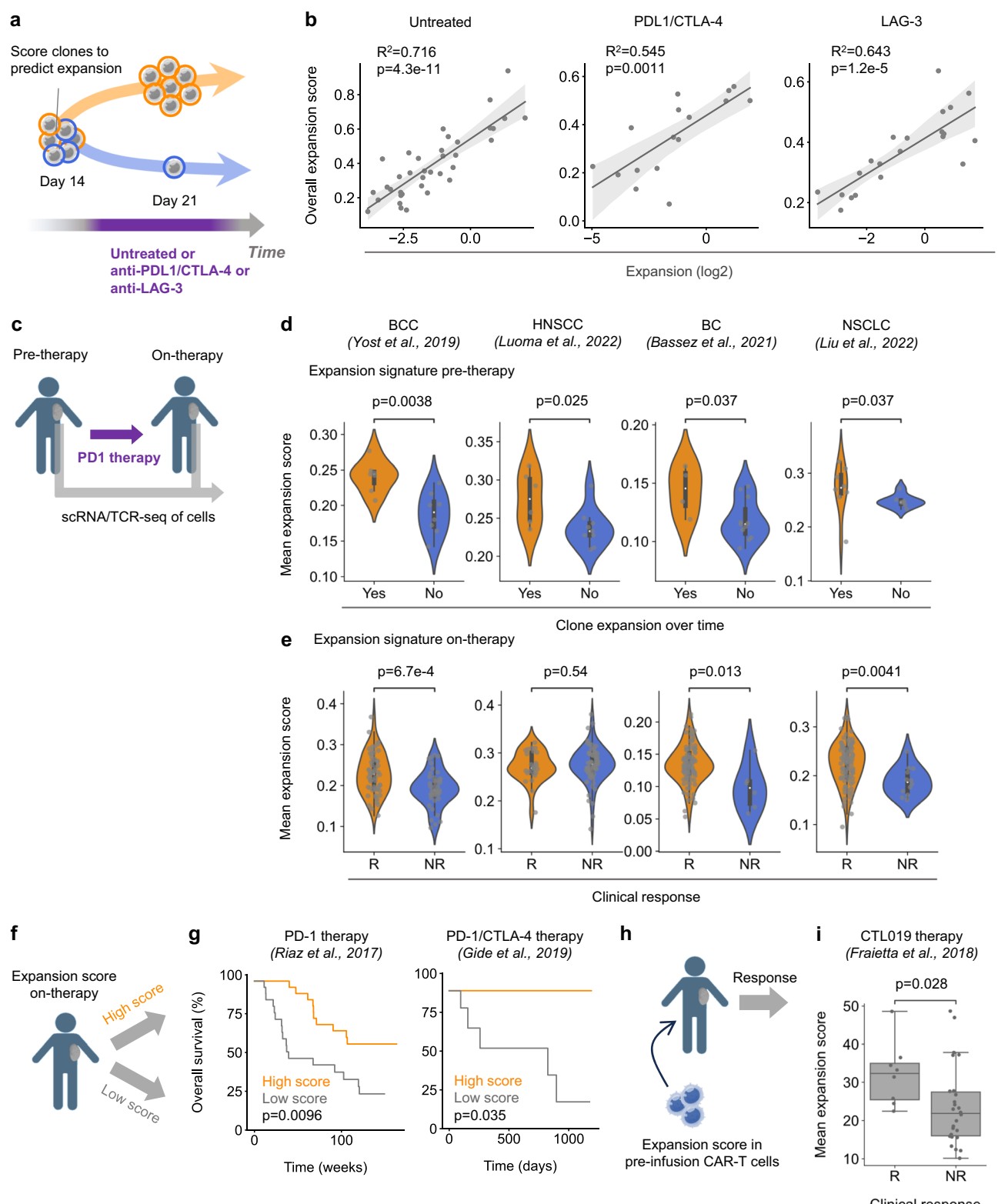

significant increase in overall survival and progression free survival compared to the "Low" cohort without splitting the cohort based on previous exposure to CTLA-4 blockade, whereas no such or less difference was seen with the TCR.strong score (Fig. 4g, Supplementary Fig. 7c, d). The "High" expansion signature score cohort demonstrated better overall survival when each treatment regimen was analyzed separately (Supplementary Fig. 7e), indicating the power of the expansion signature in stratifying different treatment regimen

outcomes. We validated that the expansion signature could stratify immunotherapy outcomes in a separate dataset of early-during-treatment (EDT) patients undergoing PD-1/PD-L1 or PD-1/PD-L1 and CTLA-4 blockade[41] (Fig. 4g, Supplementary Fig. 7f, g).

Lastly, we investigated whether the expansion signature had utility in predicting responses to CAR-T cell therapy. We scored pre-infusion CAR-T products for expression of the expansion signature and compared patient responses (Fig. 4h). In an early trial of

**Fig. 4 | The expansion signature predicts intratumoral clone expansion during immunotherapy. a, b** Clones were assigned a weighted average expansion signature score based on their gene expression across the three cell states at day 14. This score was then correlated with the degree of clonal expansion observed under untreated, anti−PD-L1/CTLA-4, and anti−LAG-3 treatment conditions. **c** Analysis of CD8[+] T cell clones from longitudinal single-cell RNA/TCR-seq tumor biopsies from basal cell carcinoma (BCC) (Yost et al.[9]), head and neck squamous cell carcinoma (HNSCC) (Luoma et al.[20]), breast cancer (BC) (Bassez et al.[12]) and non-small cell lung cancer (NSCLC) (Liu et al.[10]) patient datasets pre (pre-therapy) and post (on-therapy) anti-PD-1 therapy. **d** Comparison of the mean expansion signature score of cells in the largest (top 15) expanding and contracting clones pre-therapy from each dataset (n = 7 and 8, 6 and 9, 4 and 11, 7 and 8, expanding and contracting clones in each dataset). Analysis restricted to clones detected both pre and post therapy. **e** Comparison of the mean expansion signature score of cells in clones (top 100) from responding and non-responding patients on-therapy (n = 57 and 43, 74 and 26, 94 and 6, 88 and 12, clones from responding and non-responding patients in

each dataset). **f** Bulk RNA-seq data from tumor biopsies of melanoma patients undergoing immunotherapy were scored for expression of the expansion signature. **g** Plots of Kaplan Meier estimator for overall survival stratified by expansion signature score (above/below median) in patients treated with anti-PD-1 (Left, Riaz et al.[39], n = 50) and anti-PD-1/CTLA-4 (Right, Gide et al.[41], n = 18). **h** Pre-infusion CAR-T cell products were scored for their expression of the expansion signature. **i** Comparison of the mean expansion signature score of pre-infusion CAR-T cells between responding and non-responding patients undergoing CTL019 therapy (Fraietta et al.[42], n = 8 responding and 26 non-responding patients). Dots represent clones (**b, d, e**) and patients (**i**). Boxplots (**d, e, i**): bottom/top of box: 25th/75th percentile; upper whisker: min(max(x), Q_3 + 1.5 * IQR), lower whisker: max(min(x), Q_1 − 1.5 * IQR), center: median. Statistical testing via two-sided Kruskal-Wallis test (**d, e, i**) and two-sided Log-rank test (**g**) and two-sided Wald test (**b**). Illustrations created with cartoons from BioRender (Ueha, S. (2025) https://BioRender.com/yx373li) and Irasutoya.com (**a, c, f, h**). Source data are provided as a Source Data file.

tisagenlecleucel (CTL019)[42,43], preinfusion products expressed a significantly higher expansion score in responders than non-responders (Fig. 4i). A similar trend was confirmed in a separate dataset of tis-cel therapy[44] (Supplementary Fig. 7h). Together, these results indicate the immediate utility of the expansion signature as a powerful 'pan-immunotherapy' signature for monitoring immunotherapy outcomes.

## LAG-3 blockade enhances the expansion signature and re-expands contracting clones in the tumor

To investigate how the expansion signature changes in clones over time, we analyzed our time-resolved day 21 single-cell RNA/TCR-seq dataset (Supplementary Fig. 8a, Supplementary Data 1) from untreated mice, focusing on precursor exhausted cells from expanding and contracting clones. Compared to terminally exhausted cells, precursor exhausted cells from both expanding and contracting clones continued to express genes associated with their stem-cell like state on day 21 (Supplementary Fig. 8b). From day 14 to 21, cells from expanding clones decreased their expansion signature scores (Supplementary Fig. 8c). Conversely, cells from contracting clones increased their expansion signature scores (Supplementary Fig. 8c). These results, combined with our finding that precursor exhausted cells from contracting clones are preferentially maintained in the tumor, indicate a possibility to re-expand contracted clones by reactivation of their precursor exhausted cells.

Following the development of CTLA4 and PD-1/PD-L1 blockade, Lymphocyte activation gene-3 (LAG-3) blockade has emerged as the third checkpoint inhibitor to be approved for clinical use[6]. While the mechanisms by which it enhances the ongoing T cell response remains incompletely understood, there is evidence suggesting it could reduce TGF-β signaling[6,45]. We therefore investigated whether LAG-3 blockade could re-activate the expansion signature in precursor exhausted cells. Tumors were excised 4 days after LAG-3 blockade, and the precursor exhausted cells (subset for CD62L[-] to enrich for expanding and contracting clones−cluster 0 in Supplementary Fig. 3c) was analyzed by bulk RNA-seq (Fig. 5a). Geneset enrichment analysis (GSEA) revealed that LAG-3 blockade significantly upregulated their expansion signature, and significantly downregulated their TGF-β signaling signature (Fig. 5b, c), indicating that LAG-3 blockade was reactivating the clone expansion signature.

We employed our multi-site tumor model to assess if LAG-3 blockade did indeed increase CD8[+] T cell clone expansions. Tumors were inoculated into the left flank, right flank and left hip of mice and sequentially removed on days 14, 21 and 28 to enable the tracking of clones over three time-points. Mice were administered LAG-3 blocking antibodies or left untreated after removal of their second tumor on day 21 (Fig. 5d). This experimental schema allowed us to assess the effect of LAG-3 blockade on clonal expansion dynamics and differentiate its effect between previously expanded and contracted clones. We

analyzed clones that were newly detected on day 28, and clones that had persisted from previous timepoints. Contrary to the reported effects of PD-1/PD-L1 blockade[9–11], LAG-3 blockade did not significantly increase the fraction of newly detected clones in the tumor (Supplementary Fig. 9a). Rather, LAG-3 blockade significantly increased the fraction of large clones that were expanding (Fig. 5e). We observed a similar difference in the fraction of large clones that were contracting (Supplementary Fig. 9b). We labeled each of the large, expanding clones by their day 14 to 21 dynamics (Fig. 5f). Compared to the untreated control, many of the clones that were expanding after LAG-3 blockade had contracted between day 14 and 21. When we analyzed all clones, we observed that LAG-3 blockade significantly increased the fraction of contracted clones that were (re-)expanding by almost two-fold (Fig. 5g). Several contracted clones were re-expanding to frequencies >5 times higher than their day 21 frequencies−with one clone re-expanding 33-fold (Fig. 5h, i). These expansions altered the characteristics of CD8[+] T cells infiltrating the tumor. LAG-3 blockade increased the combined frequency of expanding clones by almost 20% (from 36.3% to 55.6%, on average) (Supplementary Fig. 9c) with most of this increase attributable to expanding clones that were stable or had contracted between day 14 and 21 (Supplementary Fig. 9d). Together, these results demonstrate that LAG-3 blockade enhances the expansion signature and re-expands contracted clones in the tumor.

## Discussion

In this study, we utilized the multi-site tumor mouse model to longitudinally assess CD8[+] T cell responses to tumors. We identified a transcriptomic gene signature that predicts the expansion of CD8[+] T cell clones. We validated this signature in other mice and human datasets and observed that LAG-3 blockade upregulates the signature and re-expands contracted clones. Developing direct methods to prospectively identify clones with high expansion signatures would help determine whether upregulation of this signature functionally contributes to clonal expansion.

Our study of the effects of LAG-3 blockade on ongoing T cell responses complement recent works investigating the role of LAG-3 in activating T cells[46,47], and, together with recent human trial data[48], provide the rationale for its potency in combination therapy[6]−with PD-1/PD-L1 blockade effectively recruiting novel clones into the tumor site[9–11], LAG-3 blockade assists by re-expanding clones already existent in the tumor. We observe the greatest effect of LAG-3 blockade on re-expanding contracting clones. TGF-β signaling is highest in contracting clones and its suppressive effect, may in part be reinforced by LAG-3. With LAG-3 blockade, the contracting clones could be preferentially relieved of one major brake, tipping their dynamics from contraction towards re-expansion. Future mechanistic studies using the multi-site tumor model could explore how immune checkpoints differentially regulate the expansion and contraction phases, guiding the

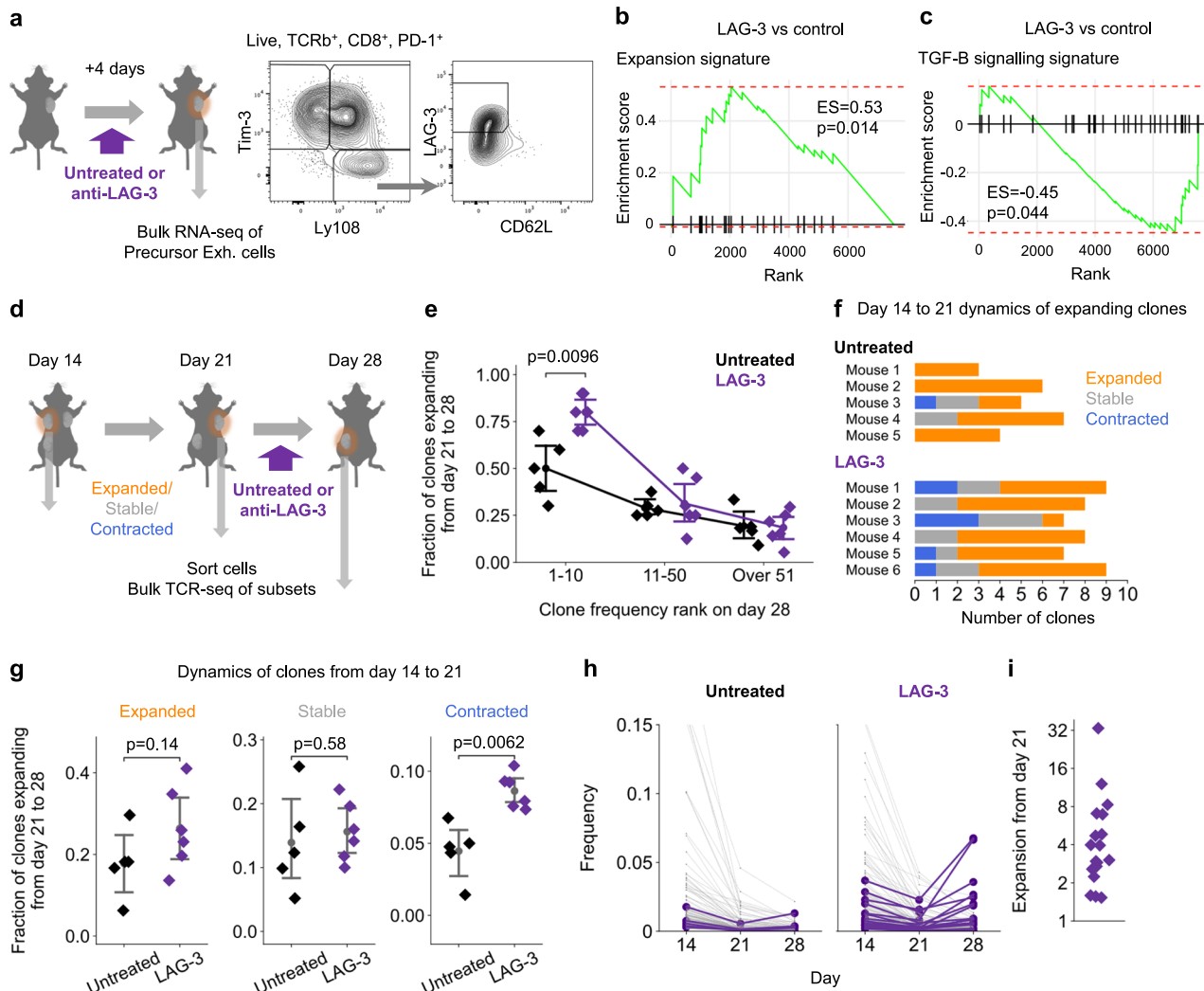

**Fig. 5 | LAG-3 blockade enhances the expansion signature and re-expands contracting clones in the tumor. a** Tumors were resected from mice on day 18, 4 days after starting anti-LAG-3 or no treatment ($n = 6$, 5 for anti-LAG-3 and no treatment respectively). **b**, **c** Geneset enrichment analysis (GSEA) testing enrichment of the expansion signature (**b**) and TGF-B signaling signature (**c**) in genes overexpressed in the precursor exhausted cells after anti-LAG3 treatment over untreated mice. Enrichment score (ES) and p-values as shown. **d** Tumors were resected sequentially from multi-site tumor mice as shown. After day 21, mice were treated with anti-LAG-3 or no treatment, ($n = 6$, 5 mice for anti-LAG-3 and no treatment respectively). Clones containing TIM-3+ cells were selected for downstream analysis. **e** Fraction of clones detected in the tumor on day 28 expanding between day 21 and 28, grouped by the ranked size of the clone on day 28 and colored by each treatment regimen ($n = 6$, 5 mice for anti-LAG-3 and no treatment respectively). **f** Expanding clones from the top 10 clones at day 28 (as compared in

(**e**)) colored by their day 14–21 expansion dynamics. **g** Fraction of clones expanding between day 21–28 for clones that expanded (Left), were stable (Middle) or contracted (Right) between day 14 and 21 ($n = 6$, 5 mice for anti-LAG-3 and no treatment respectively). **h** The change in frequency between day 14 and 28 of the largest clones (showing top 50 clones on days 14 or 21) that contracted between day 14 and 21. Clones expanding between day 21 and 28 depicted in purple ($n = 6$, 5 mice for anti-LAG-3 and no treatment respectively). **i** Expansion (the fold-increase in clone frequency from day 21 to 28) of clones from (**h**) after anti-LAG-3 treatment. Dots represent clones (**h**, **i**) and mice (**e**, **g**) with mean and 95% confidence interval as shown (**e**, **g**). Statistical testing via two-sided Kruskal-Wallis test (**e**, **g**). For statistical tests used in (**b**, **c**), see "Methods". Illustrations created with cartoons from BioRender (Ueha, S. (2025) https://BioRender.com/yx373li) (**a**, **d**). Source data are provided as a Source Data file.

development of strategies to selectively enhance phase-specific immune responses.

In our study, we demonstrate that expression of the expansion signature in biopsies taken before immunotherapy treatment can predict clone expansion during treatment. We show that expression of the expansion signature in biopsies taken during immunotherapy treatment can stratify patient outcomes. With respect to clinical responses, these results suggest that clone expansion plays an important role in determining patient outcomes after treatment has started. Other factors—such as the potential for recruitment of novel clones[9], in addition to the potential of clone expansion in pre-existing clones, may play an important role in predicting patient outcomes from biopsies taken before treatment. Interestingly, we find that

expression of the expansion signature in CAR-T cell infusion products (pre-treatment) corresponds with patient outcomes. Given that almost all immunotherapies, including CAR/TCR-T cell therapies, vaccines, and monoclonal antibodies, likely rely on CD8+ T cell expansions for their success, we envision the expansion signature could form the basis for an even more generic 'pan-immunotherapy' signature that enables accurate monitoring of patient outcomes for existing treatments, and cost-effective evaluation of immunotherapies in development.

Our results also indicate the potential of targeting the expansion signature for developing potent immunotherapies. One of the transcription factors in the expansion signature, *Tcf7*, is well-characterized as regulating the development of the precursor exhausted state[49] and serves as a biomarker for responders to immune checkpoint blockade

therapies in clinical settings[50]. Our work suggests that *Tcf7* also plays a role in regulating the expansion of clones. Targeted alterations of *Tcf7* expression, and expression of the other genes identified in the expansion signature could enable regulation of clonal expansion dynamics. These interventions would be particularly transformative for CAR/TCR-T therapy. We also find several correlations between the expansion signature and gene signatures linked to biological signaling pathways and processes, such as TGF-β signaling. These pathways could be targeted to enhance intratumoral expansion, but further experiments are required to functionally validate these correlations.

More broadly, our results indicate that future clone expansion characteristics are, at least in part, encoded in transcriptomic profiles. This leaves exciting questions to pursue—could transcriptomic profiles of CD8+ T cells encode other temporal characteristics, such as differentiation fate? Is a large component of temporal CD8+ T cell behaviors pre-determined? We posit that similar approaches combining multi-modal snapshot analyses with temporal tracking could identify transcriptomic signatures relevant for other clonal expansion dynamics, including the expansion of engrafted stem cells and the development of malignant tumors.

## Methods

### Experimental mice and cell lines
All animal experiments were conducted in accordance with institutional guidelines with the approval of the Animal Care and Use Committee of Tokyo University of Science. The maximum tumor size allowed by the institutional ethical board was 20 mm diameter or 10% of body weight, and this was not exceeded. Eight-week-old female C57BL/6 mice (CD90.2, RRID: MGI:5488963) were purchased from Sankyo Labo Service Corporation Inc. All mice were bred at specific pathogen-free facilities at Tokyo University of Science and housed under a 12-h light/dark cycle at a controlled room temperature of 23 ± 2 °C and relative humidity of 55 ± 10%. The Lewis lung carcinoma (LLC) cell line was provided by Nihonkayaku (Tokyo, Japan).

### Tumor model and treatment
LLC cells ($5 \times 10^5$ cells) were inoculated subcutaneously into the left flank, right flank and (for experiments with three tumors) the left hip of C57BL/6 J mice. In all experiments, except for control experiments to validate the proportional infiltration of clones, tumors were excised sequentially in the order of left flank tumor, right flank tumor and (for experiments with three tumors) left hip on days 14, 21 and (for experiments with three tumors) 28 or days 14 and 18 (for the bulk RNA-seq experiment). Tumor diameter was measured twice weekly and used to calculate tumor volume (mm³) [(major axis; mm) × (minor axis; mm)²]. Tumors were excised by sedating and opening a slit in the skin of the mouse, detaching the tumor from the peritoneum and the skin along with surrounding fat and draining lymph node, and clipping the slit back together. For treatment, anti-LAG-3 (clone C9B7W, BioLegend or Selleck) and combined anti-PD-L1 (clone 10 F.9G2, BioLegend) and anti-CTLA-4 (clone 9D9, Bioxcell) were injected intraperitoneally at a dose of 200 μg per mouse (100 μg per antibody per mouse for combined anti-PD-L1 and anti-CTLA-4 treatment) on days 14 and 16 (for bulk RNA-seq and scRNA/TCR-seq experiments on two tumors) or days 21 and 23 (for the bulk TCR-seq experiment on three tumors) after tumor inoculation. For the FTY720 experiment, FTY720 (Selleck) was injected intraperitoneally at a dose of 20 μg per mice on days 14, 16, 18 and 20. Mice with incomplete excision resulting in immediate (<7 days) regrowth of excised tumors, or tumors that did not grow (<100 mm³ by day 28) were excluded from the analysis.

### Generation of single-cell suspensions from tumors
Tumors were cut into small fragments and digested for 45 minutes at 37 °C with 0.2% collagenase (FUJIFILM Wako) and DNase I (Sigma-Aldrich), except for experiments for single-cell analysis, in which tumors were digested with 0.2% collagenase NB8 (Nordmark Pharma GmbH). The cells were then subjected to two-phase density separation with upper 35% Percoll PLUS (GE Healthcare) (1000 g, 10 min at room temperature). Leukocytes were recovered, washed and resuspended in preparation medium (PM) [5%FBS, 10 mM HEPES (Nacalai), Penicillin-Streptomycin (Nacalai) in RPMI-1640 (Nacalai)]. After RBC lysis with ACK buffer [155 mM of NH₄Cl (Nacalai), 10 mM of KHCO₃ (Nacalai), and 0.5 mM EDTA pH8.0 (NIPPON GENE)] for 1 min at room temperature, cells were washed and resuspended in PM, and the cell concentrations were measured using Flow-Count fluorospheres (Beckman Coulter) and a CytoFLEX S flow cytometer (Beckman Coulter).

### Cell staining and pre-enrichment
Cells were stained with a mix of Fc Block (anti-mouse CD16/CD32 mAb; clone 2.4G2, BioLegend) and fluorophore-conjugated anti-mouse monoclonal antibodies (Supplementary Table 2). Antibody mixtures were prepared using BD Horizon Brilliant Stain Buffer Plus (BD) diluted 5x with FACS buffer (D-PBS(-) supplemented with 2% FBS, and 0.05% Sodium Azide). Staining was performed for 20 min at 4 °C. In the single-cell RNA-seq/TCR-seq experiment, cells were stained with antibodies and anti-CD45 Sample Tag oligonucleotide-conjugated antibodies from the Single-Cell Multiplexing Kit (BD Biosciences). Before sorting, T cells were enriched from tumor suspensions by further staining with BD IMag APC Magnetic Particles-DM (BD Biosciences), then placed on magnetic plate. After washing by MACS buffer (D-PBS(-) supplemented with 10% BSA, and 2 mM EDTA pH8.0), APC and APC-Cy7(CD8, mouse TCRβ)-positive cells were collected as positive fraction.

### Cell sorting
Cell sorting was performed using a FACS Aria II or Aria III (BD Biosciences). Cells were stained with 5 μg/mL of propidium iodide (PI, Sigma-Aldrich) immediately before sorting. Nonviable cells were excluded from the analysis based on forward and side scatter profiles and PI staining. For bulk TCR repertoire analysis and bulk RNA-seq analysis, cells were sorted into Lysis buffer [1% w/v LiDS, 100 mM Tris-HCl pH7.5, 500 mM LiCl, 5 mM dithiothreitol, and 10 mM EDTA pH8.0] and stored at −80 °C. For single-cell RNA/TCR-seq, cells were sorted into D-PBS(-) with 10% FBS. For gating strategy, please see Supplementary Fig. 10.

### Sample collection
For the bulk TCR-seq experiments, cells were sorted from the CD8+, CD8+PD-1+Ly108+TIM-3-, CD8+PD-1+Ly108+TIM-3+, CD8+PD-1+Ly108-TIM-3+ and CD8+PD-1- fractions to produce five datasets per mouse. For the single-cell RNA/TCR-seq and bulk TCR-seq experiment, cells were sorted from the CD8+PD-1+Ly108+TIM-3-, CD8+PD-1+Ly108+TIM-3+, CD8+PD-1+Ly108-TIM-3+ and CD8+PD-1- fractions. From each of the fractions, a fixed number of cells was aliquoted for single cell sequencing, and the rest was processed by bulk TCR-seq. The numbers of cells collected for each data in each analysis are shown in Supplementary Data 1. For bulk RNA-seq samples, cells were sorted from the CD8+PD-1+Ly108+TIM-3-CD62L- fraction.

### Library preparation and sequencing for TCR repertoire analysis
TCR sequencing libraries for next-generation sequencing were prepared as previously reported with two modifications[15]. In the first and second amplifications, 18 and 10 cycles of polymerase chain reaction (PCR) were performed respectively. Oligonucleotide sequences used in this study are shown in Supplementary Table 1.

### Library preparation and sequencing for single cell analysis
Single cell sequencing libraries were prepared as previously reported[11]. Briefly, live cells were stained with Calcein AM (Nacalai) and 20,564

cells were loaded on an BD Rhapsody cartridge (BD) and processed using BD Rhapsody Cartridge Reagent Kit (BD) following the manufacturer's instructions. Single-cell cDNA synthesis was performed using BD Rhapsody cDNA Kit (BD) following the manufacturer's protocol. Whole transcriptome analysis (WTA), TCRseq, and Sample tag Index libraries were prepared using Immune Response Panel Mm (BD) according to the manufacturer's instructions with modifications[11]. Amplified libraries were quantified using a KAPA SYBR Fast qPCR Kit (KAPA Biosystems) and its size distribution was analyzed by a MultiNA. WTA, TCRseq, and Sample tag libraries were sequenced on an Illumina Novaseq 6000 S4 flowcell (67 bp read 1 and 140 bp read 2) using NovaSeq 6000 S4 Reagent Kit v1.5 to a depth of ~20,000 reads, 7,000 reads, and 10,000 reads per cell, respectively.

## Library preparation and sequencing for bulk RNA sequencing analysis

Bulk RNA-seq library preparation was performed with on-beads reverse transcription and template-switching. mRNA-trapped oligo-dT-immobilized Dynabeads M270-streptavidin (Thermo Fisher Scientific) were washed as in bulk TCR-seq library preparation, and resuspended in 10 μL of RT mix [1× First Strand buffer (Thermo Fisher Scientific), 1 mM dNTP, 2.5 mM DTT (Thermo Fisher Scientific), 1 M betaine (Sigma-Aldrich), 9 mM $MgCl_2$ (NIPPON GENE), 1 U/μL RNaseIn Plus RNase Inhibitor (Promega, Madison, WI), 10 U/μL Superscript II (Thermo Fisher Scientific), and 1 μM of trP1-TSO], and incubated for 60 min at 42 °C and immediately cooled on ice. Beads were washed once with B&W-T buffer [5 mM Tris-HCl pH 7.5, 1 M NaCl (NACALAI TESQUE), 0.5 mM EDTA, and 0.1% Tween-20 (Sigma-Aldrich)], and once with 10 mM Tris-HCl pH 8.0. To amplify the WTA library, cDNA-immobilized beads were resuspended with 25 μL of the first PCR mixture [0.4 μM of primers (illumina-i7-BCXX-trP1, and 5' BDWTA V2 primer), and 1x KAPA Hifi Hotstart ReadyMix (KAPA Biosystems, Wilmington, MA)], and amplified by PCR with the following conditions: 95 °C for 3 min, [98 °C for 20 sec, 65 °C for 30 sec, 72 °C for 5 min] x5, 72 °C for 5 min, and hold at 4 °C. The first WTA products were purified by an AMPure XP kit (Beckman Coulter, CA) at 0.7:1 ratio of beads to sample and eluted with 20 μL of 10 mM Tris-HCl pH 8.0. 10.5 μL of the first PCR products were mixed with 14.5 μL of the second PCR mixture [0.4 μM of primers (5' BDWTA V2 primer and illumina-i7-primer), and 1x KAPA Hifi Hotstart ReadyMix], and amplified with the same conditions as the first PCR. The second PCR products were purified by an AMPure XP kit at 0.7:1 ratio of beads to sample and eluted with 15 μL of 10 mM Tris-HCl pH 8.0, quantified by a Nanodrop 8000 (Thermo Fisher), and their size distribution was analyzed by a MultiNA system (Shimazu). 100 ng of the WTA library was subjected to fragmentation/end-repair/A-tailing using NEBNext Ultra II FS DNA Library Prep Kit for Illumina (New England Biolabs). Purified adapter-ligated products were subjected to index PCR. The PCR products were purified by double size-selection by using an AMPure XP kit at 0.5:1 ratio and 0.8:1 ratio (final 1.3X) of beads to sample and eluted with 12 μL of dH2O. Amplified products were analyzed by a MultiNA system. Final transcriptome libraries, with lengths of around 300 base pairs, were quantified using the KAPA Library Quantification Kit (KAPA Biosystems). Pooled libraries were sequenced by an Illumina Novaseq 6000 S4 flowcell (67 bp read 1 and 140 bp read 2) using NovaSeq 6000 S4 Reagent Kit v1.5. Oligo sequences used for this study are shown in Supplementary Table 1.

## Data processing of bulk TCR sequencing reads

Data processing of TCR-seq was performed as previously reported[11]. Briefly, reads were trimmed and filtered with Cutadapt-3.2[51] and PRINSEQ-0.20.4[52]. Filtered reads were then aligned to a reference mouse TCR V/D/J sequence registered in the international ImMunoGeneTics (IMGT) information system with MiXCR-3.0.5[53].

## Data processing of single cell sequencing reads

Data processing of single-cell RNA-seq reads was performed as previously reported[11]. Briefly, reads were trimmed with Cutadapt-2.10[51]. Filtered reads were then annotated using the Python script provided by BD Biosciences and mapped to reference RNA using Bowtie2-2.4.2[54]. Cell barcode information of each read was then added to the bowtie2-mapped BAM files and read counts of each gene in each cell barcode were counted using mawk. Resultant count data were converted to a single-cell gene-expression matrix file and filtered for valid cells based on read counts. Expression data of targeted panel genes were converted to a Seurat object[55], log normalized and scaled (regressed by the number of unique molecular identifiers per cell) using the NormalizeData and ScaleData functions respectively. Principal component analysis (PCA) was performed using RunPCA function, and enrichment of each PC was calculated using the JackStraw and ScoreJackStraw function, and PCs that were significantly enriched ($P \le 0.05$) were selected for clustering and dimensional reduction analysis. Dimensional reduction was performed using RunUMAP function with default settings. Cell clustering was performed using FindNeighbors and FindClusters (resolution = 0.6) against the significant PCs. Mapping of the dataset to the reference atlas was performed with ProjecTILs[26] using Run.ProjecTILs with default settings. Weighted Gene Network Correlation Analysis (WGCNA) was performed on precursor exhausted cells using hdWGCNA[27] following the basic tutorial pipeline with the nearest-neighbors parameter set to 10. Pseudo-time analysis was performed using Monocle 3[56] with number of nearest neighbors set to 5, and the starting node set to the precursor exhausted cell cluster node. Further downstream single cell analysis was conducted with Scanpy 1.9.1[57], including the differential gene expression analysis (rank_genes_groups).

## Data processing of bulk RNA sequencing reads

Data processing of bulk RNA-seq data was performed as previously reported[58]. Briefly, adapter trimming and quality filtering of sequencing data were performed with Cutadpat-2.10[51]. The filtered reads were mapped to reference RNA (GRCm38 release-101) using Bowtie2-2.4.2[54], and the read number of each gene was counted. Raw data was then analyzed following standard protocols using the edgeR package[59]. Genes were filtered with filterByExpr, counts normalized with calcNormFactors, dispersion characteristics computed with estimateGLMCommonDisp, estimateGLMTrendedDisp and estimateGLMTagwiseDisp, before a negative binomial generalized log-linear model was fit with glmFit. Statistical testing between groups was performed with glmLRT to rank genes before pathway analysis with fGSEA[60]. The enrichment score (ES) and p-values were calculated based on the program's default setting. We used the same gene signatures as those described in Gene Signatures Scores (Supplementary Data 3).

## Merging T cell clones across different datasets

Within the bulk TCR-seq datasets, T cell clones were determined as TCR reads with the same TCRβ V segment, D segment, J segment, and CDR3 nucleotide sequence. Within the single-cell RNA/TCR-seq datasets, T cell clones were determined as cells with the same TCRβ CDR3 nucleotide sequence. Bulk TCR-seq datasets were merged based on shared TCRβ V segment, D segment, J segment, and CDR3 nucleotide sequences while bulk TCR-seq datasets and single-cell RNA/TCR-seq datasets were merged based on shared TCRβ CDR3 nucleotide sequences. Any cells in the single-cell RNA/TCR-seq datasets that shared TCRβ CDR3 nucleotide sequences with multiple clones in the bulk TCR-seq dataset of the same mouse were excluded from the single-cell RNA/TCR-seq analysis. For the analysis of precursor exhausted cells, we further restricted the dataset to clones that were detected in the single-cell data on both days 14 and 21 and that had consistent, non-conflicting TCRα CDR3 nucleotide sequences.

## Calculation of clone features

For the bulk TCR-seq experiments, clone frequency was calculated as the fraction of reads of a clone out of the total number of reads in the CD8⁺ T cell dataset for each mouse. For the single-cell RNA/TCR-seq and bulk TCR-seq experiments, clone frequency was calculated as the fraction of combined reads of a clone in all four bulk TCR-seq datasets, out of the total combined number of reads in all four datasets for each mouse. Read counts in each dataset were corrected by the total number of cells sorted to account for cells aliquoted for single-cell RNA/TCR-seq. Expansion of clones was calculated as the log2 fold-increase in clone frequency over the indicated days. Cell counts were calculated by correcting the bulk TCR-seq obtained frequencies with flow cytometry bead derived total CD8⁺ T cell counts. The fraction of cells PD-1⁻, precursor exhausted, intermediate exhausted and terminally exhausted for a clone was calculated by dividing the read count of the clone in each of the respective CD8⁺PD-1⁻, CD8⁺PD-1⁺Ly108⁺TIM-3⁻, CD8⁺PD-1⁺Ly108⁺TIM-3⁺ and CD8⁺PD-1⁺Ly108⁻TIM-3⁺ datasets by the combined read count of the clone across all four datasets. Clones (using their TCRβ CDR3 nucleotide sequence) were defined as expanding or contracting based on differential abundance analysis using the beta-binomial model trained on the default training dataset[18]. After identification of expanding and contracting clones (Fig. 1), all subsequent analysis was restricted to clones that contained reads in the terminally exhausted (CD8⁺PD-1⁺Ly108⁻TIM-3⁺) dataset to enrich for tumor reactive clones[10].

## Gene signature scores

Gene signature scores were calculated by normalizing and taking the log of the gene counts obtained from the pre-processing step and applying the score_genes function from Scanpy with the required gene signature, as in Tirosh et al., 2016[61]. Gene signatures from Im et al.[29], Beltra et al.[30], Galleti et al.[31], Liberzon et al.[33], Nath et al.[37], Wells et al.[36] and Tsui et al.[32] were used to score cells for a stem-cell like state, Tex^prog1 and Tex^prog2 cell states, T_PEX and T_SCM/T_CM cell states, hallmark gene signatures, TGF-β signaling signature (restricted to genes with adjusted $p$-value < 0.05 and fold-change > 1), Let-7 suppression signature, and Myb-related precursor exhausted cell state signatures respectively. To avoid inflation of correlation estimates, any genes shared between the tested signature and the expansion signature were excluded from the tested signature prior to scoring. Gene signatures are listed in Supplementary Data 3. Mouse ortholog genes[62] (based on Ensembl Biomart version 87) were used when gene sets were derived from human data or needed to be converted for analysis on human data. Scores were produced for each cell, and these were averaged across all cell members of a clone to calculate mean expression scores. The overall expression of the expansion signature for each clone was approximated by calculating a weighted average of the expansion signature scores across different cell states. Specifically, we weighted the mean expansion signature score of the clone within each cell state—precursor exhausted, intermediate exhausted, and terminally exhausted/proliferating—by the proportion of clonal cells in that state, as determined from single-cell and bulk TCR-seq data.

## Analysis of published mouse dataset

Single-cell RNA/TCR-seq dataset of antigen-signaled CD8⁺ T cells infiltrating the YUMMER1.7 tumor model[38] (GSE266361) was downloaded and reprocessed as above using Scanpy with default parameters. Cells from the day 8 intraperitoneal dataset were subset from the data, log normalized, scored for gene signatures, merged with batch balanced kNN[63] and clustered (resolution = 1.5) with new neighborhood coordinates. One cluster expressing Tcf7 and Lag3 in the tumor was identified as the precursor exhausted cell cluster. Clones were defined as cells with the same TCRα and β CDR3 nucleotide sequence.

## Analysis of published human datasets

Single-cell RNA/TCR-seq data of site-matched non-lymph node tumors from Yost et al., 2019[9] (GSE123814), Luoma et al.[20] (GSE200996), Bassez et al.[12] (EGAD00001006608) and Liu et al.[10] (GSE179994), and single-cell RNA-seq data of pre-infusion tisa-cel products from Haradhvala et al.[44] (GSE197268) were downloaded and reprocessed as above using Scanpy. Cells from the CD8⁺ T cell cluster (as defined by each study) were subset from the data, log normalized and scored for the expansion gene signature. Analysis of site-matched tumors was restricted to clones that contained at least one cell from the exhausted clusters in each study ('CD8_ex' or 'CD8_ex_act' for Yost et al.[9], 'CD8_EX' for Luoma et al.[20], 'CD8_EX ' or 'CD8_EX_Proliferating' for Bassez et al.[12] and 'Tex' for Liu et al.[10]). Clones were defined as cells with the same TCRα and TCRβ CDR3 nucleotide sequence, and were defined as expanding or contracting if they increased or decreased their cell counts by >1.5-fold respectively. Bulk RNA-seq data of tumor biopsies from Riaz et al.[39] (GSE91061) and Gide et al.[41] (PRJEB23709), and pre-infusion CAR-T cell products from Fraietta et al.[42] were processed and scored for the expansion signature, and where appropriate, the TCR.strong signature (Supplementary Data 2), as described in Elliot et al.[40]. The Kaplan-Meier estimator of survival function was calculated with scikit-survival[64]. In all datasets, patient responses were defined as described in the original studies.

## Software versions

Data was analyzed using R version 4.0.3 and R packages (Seurat 4.0.4, edgeR 4.0.16, fGSEA 1.28.0, ProjecTILs 3.5.0, hdWGCNA 0.3.00, monocle3 1.4.26), Python version 3.8.6 and Python packages (jupyterlab 4.1.5, numpy 1.24.4, pandas 1.5.2, scipy 1.10.1, scanpy 1.9.1, anndata 0.7.5, scikit-survival 0.22.2, and a modified version of statannotations[65]). Figures were produced with fGSEA 1.28.0 and EnhancedVolcano 1.20.0 in R, seaborn 0.11.0 and matplotlib 3.5.0 in Python, Prism 10, Affinity Publisher 1.10.8.

## Statistical analysis

Differential gene expression on the single-cell RNA/TCR-seq data was performed using Wilcoxon rank-sum test corrected with the Benjamini−Hochberg procedure and recorded for genes with adjusted $p$-values < 0.05. Correlations were assessed using linear least-squares regression and two-tailed Wald Test. Distributions were assessed for significance using Kruskal-Wallis with Bonferroni correction where appropriate. 95% confidence intervals were calculated based on the data's bootstrap distribution. For comparison of Kaplan Meier survival curves, patients with survival data were split into "High" and "Low" cohorts based on the on-treatment gene signature scores and analysed using a Log-rank (Mantel-Cox) test.

## Reporting summary

Further information on research design is available in the Nature Portfolio Reporting Summary linked to this article.

## Data availability

The sequencing data, and processed data generated in this study have been deposited in the Gene Expression Omnibus database database under accession code GSE303232 (control data) [https://www.ncbi.nlm.nih.gov/geo/query/acc.cgi?acc=GSE303232], GSE303087 (FTY720 data) [https://www.ncbi.nlm.nih.gov/geo/query/acc.cgi?acc=GSE303087], GSE303086 (days 14 and 21 data) [https://www.ncbi.nlm.nih.gov/geo/query/acc.cgi?acc=GSE303086], GSE303233 (days 14, 21 and 28 data) [https://www.ncbi.nlm.nih.gov/geo/query/acc.cgi?acc=GSE303233], GSE304629 (scRNA/TCR-seq untreated data) [https://www.ncbi.nlm.nih.gov/geo/query/acc.cgi?acc=GSE304629], GSE304763 (scRNA/TCR-seq immunotherapy data) [https://www.ncbi.nlm.nih.gov/geo/query/acc.cgi?acc=GSE304763] and GSE303089

(bulk RNA-seq data) [https://www.ncbi.nlm.nih.gov/geo/query/acc.cgi?acc=GSE303089]. The processed data used in this study are also available on Zenodo [https://doi.org/10.5281/zenodo.16248640]. The publicly available data used in this study are available as follows: Takahashi et al.[38] (GSE266361 [https://www.ncbi.nlm.nih.gov/geo/query/acc.cgi?acc=GSE266361]), Yost et al.[9] (GSE123814 [https://www.ncbi.nlm.nih.gov/geo/query/acc.cgi?acc=GSE123814]), Luoma et al.[20] (GSE200996 [https://www.ncbi.nlm.nih.gov/geo/query/acc.cgi?acc=GSE200996]), Bassez et al.[12] (EGAD00001006608 [https://ega-archive.org/datasets/EGAD00001006608]), Liu et al.[10] (GSE179994 [https://www.ncbi.nlm.nih.gov/geo/query/acc.cgi?acc=GSE179994]), Haradhvala et al.[44] (GSE197268 [https://www.ncbi.nlm.nih.gov/geo/query/acc.cgi?acc=GSE197268]), Riaz et al.[39] (GSE91061 [https://www.ncbi.nlm.nih.gov/geo/query/acc.cgi?acc=GSE91061]) and Gide et al.[41] (PRJEB23709 [https://www.ebi.ac.uk/ena/browser/view/PRJEB23709]). The remaining data are available within the Article, Supplementary Information or Source Data file. Source data are provided with this paper.

## Code availability

No new algorithms were developed for this manuscript. Code generated for display of each figure, and corresponding preprocessed data is deposited on Zenodo [https://doi.org/10.5281/zenodo.16248640]. Additional code generated for analysis is available from the corresponding authors upon reasonable request.

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

## Acknowledgements

This work was supported by the Japan Society for the Promotion of Science under grants 20H03474 and 23K27397 (S.U.), and by the Japan Agency for Medical Research and Development under grants JP22fk0310509 and JP25fk0310531 (S.U.), and JP22ama221306 (K.M.). J.E.D.T. was funded by the Medical Research Council (MC_UU_0025/12) award. D.B. was funded by the Medical Research Council (MR/V009052/1) and a Lister Institute of Preventive Medicine fellowship. M.T. was sup-ported by the Masason Foundation, Recruit Foundation and the Gates Cambridge Trust. We thank Y. Hara (Tokyo University of Science) for advice regarding cell sorting; members of IGT, Inc. for expert technical assistance with TCR sequencing; N. Yamaguchi (Washington University in St Louis), M. Nakano (University College London) and Y. Kawamura (Uni-versity of Cambridge) for helpful discussions. Illustrations in Figs. 1a, 1c, 2a, 2e, 3b, 4a, 4c, 4f, 4h, 5a, 5d and Supplementary Fig. 6a were created with cartoons from BioRender.com and Irasutoya.com.

## Author contributions

M. Takahashi, M. Tsunoda, H.A. and S.U. conceived the project and designed the experiments. M. Tsunoda performed the experiments with help from H.A., M.K., H.O. and S.U. M. Takahashi analyzed the data with help from H.A., S.S., D.B. and M. Tsunoda. S.U. and K.M. supervised the project with support from J.E.D.T. and I.S. M. Takahashi, M. Tsunoda and S.U. wrote the manuscript. All authors reviewed the results and approved the final version of the manuscript.

## Competing interests

Author M.Tsunoda is currently an employee of Daiichi Sankyo Company, Limited. M.Tsunoda was a student at Tokyo University of Science at the time of this study and Daiichi Sankyo Company, Limited was not involved in this study. S.S. reports an advisory role for ImmunoGeneTeqs, Inc. and stocks for ImmunoGeneTeqs, Inc. K.M. reports a consulting or advisory role for Kyowa-Hakko Kirin and ImmunoGeneTeqs, Inc; research funding from Kyowa-Hakko Kirin and Ono; and stocks for ImmunoGeneTeqs, Inc. and IDAC Theranostics, Inc. S.U. reports an advisory role for Immuno-GeneTeqs, Inc. and stocks for ImmunoGeneTeqs, Inc. and IDAC Ther-anostics, Inc. The remaining authors declare no competing interests.
