## [Transparent Peer review file · Nature Communications]

A pan-immunotherapy signature to predict intratumoral CD8⁺ T cell expansions

Corresponding Author: Dr Satoshi Ueha

Version 0:

Reviewer comments:

Reviewer #1

(Remarks to the Author)

In the manuscript titled A pan-immunotherapy signature to predict intratumoral CD8⁺ T cell expansions, the authors used an elegant multi-tumor mouse model approach to longitudinally assess CD8⁺ T cell responses to tumors. They claimed to discover a transcriptomic gene signature that predicts the expansion of CD8⁺ T cell clones. They pursued the validation of this signature in other mice and human datasets and observed that LAG-3 blockade upregulates it and re-expands contracted clones.

The approach is very elegant, and the findings are very well presented and described, though I respectfully consider that extra work is necessary to be formally published in this prestigious journal.

1-In Extended Data Fig. 1a-b the authors described a PD-1⁺ Ly108⁺TIM-3⁺. This subset was a bit weird to me, since Ly108 is always perfectly correlated with TCF1 and PD-1⁺ TCF1⁺ Tpex cells never express TIM-3. Could the authors develop on this, because intermediate exhausted do not express TIM-3 and are Ly108-CD69- (Beltra et. al <https://doi.org/10.1016/j.immuni.2020.04.014>)?

2-The authors reasoned that comparison of clones across sequentially removed tumors could therefore enable the tracking of clones over time (Fig. 1c). This is a very fair assumption, but is only truly valid if surgical excisions did not influence the clonal dynamic in remained tumors. So, in what manner did the authors assess the quality control of the surgical procedure, specifically to rule out that it did not influence the clonal dynamic in the leftover tumor through antigen spreading? For instance, an experiment to assess clonal dynamic involving surgery plus/minus FTY720 could help to address this issue.

3-Fig.2 clearly shows that clonal expansion is mostly associated with an active Tpex-*Tex* differentiation process, where both compartments increased in size for most of the expanding clones (Fig.2c, upper right quarter). Indeed, some contracting clones increased in Tpex-like cells, Fig.2c, lower right quarter), confirming that clonal expansion is not only associated with the capabilities of Tpex cells to expand, Tpex-derived *Tex* cells should also be generated and proliferate. I wonder whether it is also possible to identify a transcriptomic signature in *Tex*-like cells that can predict clonal expansion? Do the authors think that this should be an exclusive property of Tpex cells? If so, why?

4- Can the authors assess the correlation between the expansion signature and the signature derived from Myb⁺ Tpex (<https://doi.org/10.1038/s41586-022-05105-1>). Indeed, here is shown that the proliferative burst in response to PD-1 checkpoint inhibition originates exclusively from CD62L⁺ TPEX cells.

5-Why do exist two separate cluster of Tpex cells in Fig.3a? this should be not related with the capability to expand since from Fig3c is clear that these cells are almost identical.

6-The authors commented that comparing responders and non-responders, clones from responders in post-treatment samples had significantly higher expansion signature scores than clones from non-responders, while no such difference was observed in pre-treatment samples (Fig. 4b, Extended Data Fig. 5c). However, that latter should be also expected since

day 14 TILs displayed higher expansion score than day 21 TILs (Extended Data Fig. 6c). Conversely, cells from contracting clones increased their expansion signature scores (Extended Data Fig. 6c). So to what extent this score is a real predictor of clone that will undergo significant expansion overtime.

7-Could this expansion signature stratify TIL-ACT products based on expansion/clinical response post cell transfer? For this, the authors can use the dataset from Krishna et.al (DOI: 10.1126/science.abb9847) and also response to PD-1 blockade in Bassez, A. et al (<https://doi.org/10.1038/s41591-021-01323-8>)

8- From Fig.5 it seems very clear that the LAG3 is restringing the upregulation of the expansion signature. Therefore, I recommend the following experiment to formally demonstrate that this signature is associated with expansion.

Following the experimental set up of Fig.5a, I recommended to sort Lag3+ and Lag3- Pex cells, transfer them to new tumor-bearing host and treat both cohorts with/ PD-1 Blockade/control Ab and follow up expansion and tumor control, please see (<https://doi.org/10.1038/s41590-019-0312-6>)

9- Do the expansion and tumor control is better upon combining LAG3- and TGFbeta blockade?

Reviewer #2

(Remarks to the Author)

In this study, Takahashi et al. perform a time-resolved analysis of clonal T cell dynamics in preclinical mouse tumor models. The experimental approach, combinign a multi-site tumor model with mapping of clonal T cell responses by single-cell RNA-sequencing and TCR-sequencing, is elegant. Several of the findings on the dynamics of T cell expansion and contraction in tumor-bearing mice are interesting for our understanding of the dynamics of clonal T cell responses in general. However, what remains unclear is to which extent the authors T_{pex} expansion signature, identified in a non-treatment setting, can truly predict T cell expansion to cancer (pan-) immunotherapy. This main message of the paper remains to be thoroughly addressed to improve the manuscript and warrant its publication in Nature Communications.

Major comments

1. In their LLC mouse cancer model, the authors identify a gene expression signature in T_{pex} that is (1) linked to the future expansion of T cell clones in absence of immune checkpoint blockade (Figure 3) and (2) upregulated in T cells expanding after anti-LAG-3 treatment in the same model (Figure 5). But can its analysis in T_{pex} before anti-LAG-3 treatment be used to predict clonal T cell expansion past anti-LAG3 treatment ? It is puzzling that the authors did not assess this, as the information should be contained in their datasets shown in Figure 5. This important point needs to be clarified to justify the authors claim that their signature is predictive of intratumoral T cell expansion across different immunotherapies, this important point needs to be a) clarified in the data from experiments with anti-LAG-3 treatment and b) extended to experiments with PD-1/CTLA4 blockade.

2. Related to the above: the authors do not find a pre-treatment enrichment of their expansion signature in human melanoma patients responding to anti-PD1/PD-L1 blockade (Extended data Figure 5d). This suggests that the signature, although as shown in Figure 4b, is induced by succesfull ICB treatment, does not allow to predict immunotherapy response in these cancer patients – These data argues for only limited power of the predictive signature in human cancer patients. They further are inconsistent with the authors findings for the basal cell carcinoma dataset shown in Figure 4a. Given these discrepancies, it seems important that he overall point of whether or not expression of the expansion signature in T_{pex} can or cannot predict immunotherapy remains unclear – and should be clarified by probing other scRNA-Seq datasets from human cancer patients to evaluate its predictive power.

3. The findings by the authours indicate that TGF-beta signaling may determine the capacity of T_{pex} cells to expand in tumors, which seems interesting with respect to the mechanisms limiting T_{pex} responses in tumors. Can the authors demonstrate in loss-of-function or gain-of-function experiments that TGF-beta impacts clonal T cell expansion in tumors?

Minor comments

4. Tsui et al., Nature 2022, have reported on CD62L+ T_{pex} driving clonal expansion. Is a signature for CD62L+ T_{pex} correlated with clonal T cell expansion in the authors dataset ?

Reviewer #3

(Remarks to the Author)

Major comment

(1) If I understand correctly, when the authors did the experiments, the expanding clones and contracting clones were taken from the same 2 snapshots of times in the same mice, right? That means they are independent clones that occur simultaneously. But in Fig. 2e, the authors draw the plot in a way to indicate that T cell clones first expand and then contract in a sequential manner. There doesn't seem to be evidence to support this right? Why not contract first and then expand? As I mentioned, the contracting and expanding clones are different. The authors should conduct experiments at 3 instead of 2 time points, to define the expanding/contracting history of the same clone?

(2) "These cells could not be differentiated based on 24 their Uniform Manifold Approximation and Projection (UMAP) representation (Fig. 3c, Extended 25 Data Fig. 3d). However, differential gene expression (DEG) analysis between precursor exhausted 26 cells from expanding and contracting clones revealed that cells from expanding clones 1 overexpressed 45 genes – of which 22 were canonical genes" UMAP cannot tell any difference, and there are only 22 genes that are differentially expressed. I worry the authors are reading too much into the data and the expansion signature is not very real.

(3) "50 gene signatures of well-defined biological processes, and searched for correlations with the expansion 20 signature score at the clonal level." If direct differential gene expression analyses only revealed 22 differentially expressed genes, how is it possible that the authors end up with so many hallmark pathways that are correlated? In other words, why were the genes in these hallmark pathways not found in the most direct differential gene expression analyses? This raises a concern of whether the observed correlations in this section of the result are simply a result of aggregation of random noises rather than true signal

(4) "The expansion signature stratifies melanoma patient outcomes to PD-1 and PD-1/CTLA-4 9 blockade." I feel the authors need to find at least one/ideally two other cohorts with similar data and repeat these analyses. See if the conclusions are robust

(5) "Together, these results demonstrate that LAG-3 blockade 14 enhances the expansion signature and re-expands contracted clones in the tumor." I might have missed this. But I don't seem to get why LAG-3 blockade would preferentially re-expand contracting clones as opposed to expanding clones. Could the authors explain, in the context of their theory?

Minor comments

(1) "the sizes of any given clone (as identified by the read 19 frequency of a TCR sequence in the repertoire) were equivalent" I agree the TCR frequencies are highly correlated. But "equivalent" is a big exaggeration.

(2) "We observed a consistent trend in the fraction of large clones 3 that were contracting (Extended Data Fig. 7b)." "Consistent trend" seems somewhat confusing. Will "similar fraction" or something like this be more accurate and straightforward? Is this what the authors mean?

Version 1:

Reviewer comments:

Reviewer #1

(Remarks to the Author)

The reviewer express satisfaction with the revision. The authors have effectively addressed the majority of my primary concerns. I have a few minor suggestions for their consideration:

a) Relative to my Q1, In addition to the projection, I would recommend to calculate a pseudo-time along the T_{pex}-T_{ex} differentiation trajectory, to better illustrate the potential transitional nature of the PD-1+TIM-3+ Ly108+ CD8+ TILs

b) Clarify whether the expansion signature is more enriched in proliferating vs. non-proliferating T_{ex}_terms (I did not see this clarification in the text of as part of the answer to Q3)

Reviewer #2

(Remarks to the Author)

The authors have successfully addressed all my concerns raised at first review. The new data in the revised paper significantly improve the manuscript which to me now is acceptable for publication. I congratulate the authors on a very interesting and thorough study.

Reviewer #3

(Remarks to the Author)

The authors have addressed all my concerns successfully. Thank you!

[name redacted]

REVIEWER COMMENTS

Reviewer #1 (Remarks to the Author): with expertise in cancer immunology

In the manuscript titled A pan-immunotherapy signature to predict intratumoral CD8+ T cell expansions, the authors used an elegant multi-tumor mouse model approach to longitudinally assess CD8+ T cell responses to tumors. They claimed discover a transcriptomic gene signature that predicts the expansion of CD8+ T cell clones. The pursued the validation of this signature in other mice and human datasets and observed that LAG-3 blockade upregulates it and re-expands contracted clones.

The approach is very elegant, and the findings are very well presented and described, though I respectfully consider that extra work is necessary to be formally published in this prestigious journal.

We thank the reviewer for their positive comments on our manuscript. We appreciate that the reviewer found our approach elegant, and our findings well-presented in the manuscript. We also thank the reviewers for raising multiple points that have strengthened our manuscript. We have addressed all these points, as detailed below.

1-In Extended Data Fig. 1a-b the authors described a PD-1+ Ly108+TIM-3+. This subset was a bit weird to me, since Ly108 is always perfectly correlated with TCF1 and PD-1+ TCF1+ Tpex cells never express TIM-3. Could the authors develop on this, because intermediate exhausted do not express TIM-3 and are Ly108-CD69- (Beltra et. al <https://doi.org/10.1016/j.immuni.2020.04.014>)?

We thank the reviewer for raising this important point. We agree with the reviewer that Ly108 is correlated with TCF1 expression and PD-1+ TCF1+ Tpex cells generally do not express TIM-3. In solid tumors, we and others find a population of PD-1+ Ly108+ TIM-3+ cells that are transitional states between PD-1+ Ly108+ TIM-3- precursor exhausted and PD-1+ Ly108- TIM-3+ terminally exhausted states – hence ‘intermediate exhausted’ cells¹. This population has been described previously with the same markers in preclinical mouse models (Miller et al., Nature Immunology 2019² - Fig. 4e), and similar intermediate exhausted states have been described in human patients³⁻⁵.

To better characterize these cells, we projected the clusters in our scRNA/TCR-seq datasets to a reference atlas of tumor-infiltrating T cells (ProjecTILs atlas; Andreatta et al., Nature Communications 2021⁶). The mapping shows that these intermediate cells are a transitional state between precursor exhausted and terminally exhausted cells (Fig. A).

A – Projection of clusters to the ProjectTILs reference atlas⁶. (Extended Data Fig. 3a in paper)

For better readability and interpretation of the data, we have added these analyses for each of our scRNA-seq datasets and included corresponding descriptions in our manuscript on Extended Data Fig 3a and Extended Data Fig. 6f, and Pages 6-7: Lines 25-3 as below.

Unsupervised clustering of the single-cell RNA/TCR-seq data retrieved precursor exhausted (clusters 0, 1 and 4), intermediate exhausted (clusters 2, 5, 7 and 8), terminally exhausted (cluster 3) and proliferating cells (cluster 6 – of which the majority were terminally exhausted cells) (Fig. 3a, Extended Data Fig. 3b-c), which we confirmed by projection on a reference atlas of tumor-infiltrating T cells⁶.

2-The authors reasoned that comparison of clones across sequentially removed tumors could therefore enable the tracking of clones over time (Fig. 1c). This is a very fair assumption, but is only truly valid if surgical excisions did not influence the clonal dynamic in remained tumors. So, in what manner did the authors assess the quality control of the surgical procedure, specifically to rule out that it did not influence the clonal dynamic in the leftover tumor through antigen spreading? For instance, a experiment to access clonal dynamic involving surgery plus/minus FTY720 could help to address this issue.

We thank the reviewer for raising this important point and for suggesting a simple yet powerful experiment to address this issue. We have followed the reviewer’s suggestion and undertaken an additional experiment with our multi-site tumor mouse model involving surgery plus/minus FTY720⁷. Specifically, we inoculated mice with two lateral flank tumors, removed one tumor on day 14 by surgical resection, administered FTY720 intraperitoneally at 20µg/mouse on days 14, 16, 18 and 20, and removed the remaining tumor on day 21 (Fig. B). We processed cells from the tumors by bulk TCR-seq and compared if administration of FTY720 affected the clonal dynamics in the tumor. If the surgical excision influenced the clonal dynamics in the leftover tumor through antigen spreading, we would expect to see altered clonal dynamics (such as a decrease in expanding clones) with FTY720.

Following the rest of our analysis, we focused on the ‘relevant clones’ – clones with detectable TIM-3+ reads on day 14 or 21. We found that surgery plus/minus FTY720 showed the same clonal dynamics. Specifically, there was no significant difference in the number and frequency of expanding and contracting clones. These results indicate that surgical excision does not influence the clonal dynamics in the leftover tumor through antigen spreading and provide a quality control of the surgical procedure.

B – Experimental schema to evaluate the clonal dynamics involving surgery plus/minus FTY720.

C – The number (Left), the combined frequency on day 14 (Middle) and the combined frequency on day 21 (Right) of expanding and contracting clones from tumors resected simultaneously with (Dark grey, n=7) and without FTY720 between days 14 and 21 (Light grey, n=7). (Extended Data Fig. 2b in paper)

We thank the reviewer for proposing this important experiment. We have added the results of this experiment, and corresponding descriptions in our manuscript on Extended Data Fig. 2b and Page 5: Lines 23-26 as below.

First, to assess whether sequential tumor removal influenced the clonal dynamics of tumor-specific T cells in the remaining tumor through antigen spreading, we blocked lymphocyte migration using FTY720⁷ following resection of the first tumor. FTY720 blockade had no effect on the fraction or frequency of expanding and contracting clones (Extended Data Fig. 2b).

3-Fig.2 clearly shows that clonal expansion is mostly associated with an active T_{pex}-T_{ex} differentiation process, where both compartments increased in size for most of the expanding clones (Fig.2c, upper right quarter). Indeed, some contracting clones increased in T_{pex}-like cells, Fig.2c, lower right quarter), confirming that clonal expansion is not only associated with the capabilities of T_{pex} cells to expand, T_{pex}-derived T_{ex} cells should also be generated and proliferate. I wonder whether is also possible to identify a transcriptomic signature in T_{ex}-like cells that can predict clonal expansion? Do the authors think that this should be an exclusive property of T_{pex} cells ? If so, why?

We thank the reviewer for raising this thoughtful point. We initially focused our analysis on T_{pex} cells as a large body of the literature has reported T_{pex} cells as driving responses to immunotherapy^{1,2,8-11},

and in our dataset, expanding clones had the highest fraction of Tpex cells (Extended Data Fig. 2c-d). However, we agree with the reviewer that Tpex-derived Tex cells are also generated, and could proliferate, and so the expansion signature may not be an exclusive property of Tpex cells. We therefore assessed whether (1) it is possible to identify a similar transcriptomic signature in Tex-like cells, and whether (2) the expansion signature derived from Tpex cells could also be applied to Tex-like cells.

First, we performed a differential gene analysis on Tex (terminally exhausted) cells from expanding and contracting clones. We found only one gene *lfi27l2a* differentially expressed (upregulated in cells from expanding clones). However, it is important to note that our dataset is enriched for Tpex cells, so contains only limited number of Tex cells which may limit the power of our analysis.

Next, we applied the expansion signature to the other Tex subsets: intermediate exhausted and terminally exhausted/ proliferating (as these were predominantly terminally exhausted) cells and evaluated the correlation of cells in these states with their future expansion rates. Grouping cells into clones and comparing their values against a continuous metric (expansion) could aggregate weaker signals that would otherwise remain undetected with direct comparison of expanding and contracting cells. Strikingly, we found that the expansion signature correlated with future expansion rate in each of the subsets (Fig. D). Given these results, we tested whether the overall expression of the expansion signature in clones (approximated by weighting each cell state's expansion signature score by the proportion of clonal cells in the state for each clone) could predict expansion (Fig. E). We found that the overall expansion signature score correlated strongly with expansion (Fig. F). We thank the reviewer for raising this thoughtful point - most mice and human datasets are un-enriched for particular cell states and so demonstrating that the expansion signature from cells of the 'whole' clone predicts expansion adds significant value to our manuscript.

D – Scatter plots comparing the mean expansion signature score of intermediate exhausted (Left) and terminally exhausted/ proliferating cells (Right) with the expansion of clones. Analysis on clones with at least 5 corresponding cells on day 14. (Extended Data Fig. 4g in paper)

E – Clones were assigned a weighted average expansion signature score based on their gene expression across the three cell states (precursor exhausted, intermediate exhausted and terminally exhausted/ proliferating) at day 14.

F – The overall expansion signature score was then correlated with the degree of clonal expansion observed. (Fig. 4b in paper)

We thank the reviewer for raising this important point. We have now added the results of this additional analysis, and corresponding descriptions in our manuscript on Fig 4b and Extended Data Fig. 4g, and Page 8: Lines 13-15, and Page 9: Lines 18-24 as below.

Expression of the expansion signature in in both intermediate exhausted cells and terminally exhausted/proliferating cells, however, correlated strongly with clonal expansion (Extended Data Fig. 4g).

To approximate how clonal expression profiles would appear in non-enriched or human datasets, we calculated each clone's overall expansion signature score as the weighted average of its state-specific scores, using the proportion of clonal cells in each state on day 14 as weights. Strikingly, in untreated ... conditions, the clones' expansion signature score on day 14 correlated strongly with their future expansion (Fig. 4b)...

4- Can the authors assess the correlation between the expansion signature and the signature derived from Myb+ Tpex (<https://doi.org/10.1038/s41586-022-05105-1>). Indeed, here is shown that the proliferative burst in response to PD-1 checkpoint inhibition originates exclusively from CD62L+ TPEX cells.

We thank the reviewer for suggesting this additional analysis. We agree with the reviewer that analysis with the genesets derived in this paper would be of significant interest for the field. We have therefore assessed the correlation between the expansion signature and the signature derived from Myb+ Tpex cells, specifically the genesets derived from comparing CD62L+ with CD62L- Tpex cells, and the genesets derived from comparing CD62L- with Myb KO mice Tpex cells (Fig. G). We do not observe a consistent correlation between the expansion signature score and the Myb+ Tpex cells. We observe a weak correlation between the expansion signature score and the CD62L- Tpex cell signature. The Myb+ Tpex signatures were generated from mice with chronic infections, where the effector site, and consequently the sampling site, is predominantly the spleen. This could make interpretations of gene signatures in the tumor difficult, particularly for transcriptomic signatures that may be affected by tissue location. For instance, in the tumor, we find that the majority of CD62L+ Tpex cells are from 'non-relevant' clones (clones with no TIM-3+ reads on either day 14 or 21) – indeed, projection of this cluster onto the ProjectTIL atlas⁶ suggests that these cells are likely naïve or bystander cells (cluster 4; Fig. H-K).

G – Scatter plots comparing the mean signature score of gene signatures from Tsui et al., 2022¹⁰ with the expansion of clones. Analysis on clones with at least 5 corresponding cells on day 14. (Extended Data Fig. 4f in paper)

H – UMAP map of cells colored by clusters obtained from unsupervised Louvain clustering.

I – Dot plot showing the expression of marker genes for each cluster – CD62L is enriched in cluster 4.

J – UMAP map of cells colored by whether they belong to ‘relevant clones’ (Clones with at least one read in the TIM-3 compartment at either day 14 or 21).

K – Projection of cluster 4 to the ProjecTILs reference atlas⁶

The analysis of these signatures is of significant interest for the field, and so we have added these results, and corresponding descriptions in our manuscript on Extended Data Fig. 4f, and Pages 8: Lines 10-12 as below.

Other previously reported precursor exhausted cell state gene signatures from mice and humans²⁹⁻³², including from a recently described CD62L+ precursor exhausted population³², did not consistently correlate with the expansion of clones (Extended Data Fig. 4e-f).

5-Why do exist two separate cluster of T_{pex} cells in Fig.3a? this should be not related with the capability to expand since from Fig3c is clear that these cells are almost identical.

We thank the reviewer for highlighting this confusing figure. As we describe in our response to the previous point (Point 4), one of the clusters in Fig. 3a predominantly contains CD62L+ T_{pex} cells that are not members of ‘relevant clones’ (clones with no TIM-3+ reads on either day 14 or 21) – projection of this cluster onto the ProjecTIL atlas⁶ suggests that these cells are likely naïve or bystander cells (cluster 4; Fig. H-K). We agree with the reviewer that the current display of the figure is confusing. The reviewer’s point also highlights that the figure is inconsistent with the rest of the figures in the paper where we have exclusively focused on the ‘relevant clones’. We apologies for this oversight and have accordingly remade the UMAP figure to be consistent with the other figures by filtering for ‘relevant clones’ before dimensionality reduction (L-N).

Cells from bilateral tumors excised on days 14 and 21 from 3 mice were processed by a combination of single-cell RNA/TCR-sequencing and bulk TCR-sequencing to enable integration of single-cell transcriptomic data of clones with their expansion dynamics.

L – Uniform Manifold Approximation and Projection (UMAP) map of cells colored by clusters obtained from unsupervised Louvain clustering. (Extended Data Fig. 3b in paper)

M – UMAP map of 2928 cells from tumors excised on day 14, colored and labelled by cell type. (Fig. 3a in paper)

N – UMAP map of precursor exhausted cells colored by their expansion dynamics. (Fig. 3c in paper)

6-The authors commented that comparing responders and non-responders, clones from responders in post-treatment samples had significantly higher expansion signature scores than clones from non-responders, while no such difference was observed in pre-treatment samples (Fig. 4b, Extended Data Fig. 5c). However, that latter should be also expected since day 14 TILs displayed higher expansion score than day 21 TILs (Extended Data Fig. 6c). Conversely, cells from contracting clones increased their

expansion signature scores (Extended Data Fig. 6c). So to what extent this score is a real predictor of clone that will undergo significant expansion overtime.

We thank the reviewer for raising this important point. We understand the reviewer's concern regarding to what extent this score is a real predictor of clone that will undergo significant expansion overtime. We have therefore generated two additional scRNA/TCR-seq datasets and conducted additional analysis on third-party human datasets to demonstrate that the expansion signature predicts clones that will undergo significant expansion overtime in both mice and human datasets.

First, we generated two additional time-resolved scRNA/TCR-seq dataset using our multi-site tumor model. We collected transcriptomic/ bulk TCR-seq data on day 14, and bulk TCR-seq data on day 21, while the mice were treated with anti-LAG-3 (200 $\mu\text{g}/\text{dose}/\text{mouse}$) or anti-PD-1/CTLA-4 (100 μg each/ dose/ mouse) blockade intraperitoneally on days 14 and 16. This new dataset enabled us to match the transcriptomic states of single CD8⁺ T cells before therapy, with their clonal expansion dynamics after anti-LAG-3 or anti-PD-1/CTLA-4 therapy. We used this dataset to evaluate whether the expansion signature correlates with clone expansion in separate murine datasets under immunotherapy. In line with our response to Point 3, we tested whether the overall expression of the expansion signature in clones (approximated by weighting each cell state's expansion signature score by the proportion of clonal cells in the state for each clone) could predict clone expansion (Fig. O). The overall expansion signature score before treatment strongly correlated with expansion during treatment in both the new datasets (under both anti-PDL1/CTLA-4 and anti-LAG-3 blockade) (Fig. P), demonstrating that the expansion signature predicts clones that undergo significant expansion in immunotherapy datasets.

O – Clones were assigned a weighted average expansion signature score based on their gene expression across the three cell states at day 14. This score was then correlated with the degree of clonal expansion observed under new datasets: anti-PD-L1/CTLA-4, and anti-LAG-3 treatment conditions. (Fig. 4a in paper)

P – Scatter plots comparing the overall expansion signature score with the expansion of clones in the new datasets: anti-PDL1/CTLA4 (Left) and anti-LAG-3 (Right) treatment. Analysis on clones with at least 3 cells in each cell state on day 14. (Fig. 4b in paper)

We thank the reviewer for suggesting we clarify this important point. We have added the results of this analysis, and corresponding descriptions in our manuscript on Fig 4a and 4b, and Page 9: Lines 10-24 as below.

To establish the utility of the expansion signature, especially in the context of immunotherapy, we tested whether clonal expansion could be predicted from the expansion signature in both untreated and immunotherapy treated conditions (Fig. 4a). Using the multi-site tumor model, we generated additional time-resolved scRNA/TCR-seq datasets to track clonal responses under anti-PDL1/CTLA4 and anti-LAG-3 blockade. The new datasets enabled us to match the transcriptomic states of single CD8⁺ T cells before therapy, with their clonal expansion dynamics after therapy (Extended Data Fig. 6a). As before, unsupervised clustering of the single-cell RNA/TCR-seq data retrieved precursor exhausted, intermediate exhausted, terminally exhausted and proliferating cells (of which the majority were terminally exhausted cells) (Extended Data Fig. 6b-f). To approximate how clonal expansion profiles would appear in non-enriched or human datasets, we calculated each clone's overall expansion signature score as the weighted average of its state-specific scores, using the proportion of clonal cells in each state on day 14 as weights. Strikingly, in untreated and in both anti-PDL-1/CTLA-4 and LAG-3 conditions, the clones' expansion signature score on day 14 correlated strongly with their future expansion (Fig. 4b), indicating that the expansion signature could be used to predict clone expansion during immunotherapy.

Next, we tested our expansion signature on additional patient samples. We analyzed site-matched scRNA/TCR-seq biopsies taken from patients pre- and post-PD-1 blockade therapy. The datasets consisted of biopsies taken from basal cell carcinoma (BCC) (Yost et al., *Nature Medicine* 2019⁴), head and neck squamous cell carcinoma (HNSCC) (Luoma et al., *Cell* 2022¹²), breast cancer (BC) (Bassez et al., *Nature Medicine* 2021¹³) and non-small cell lung cancer (NSCLC) (Liu et al., *Nature Cancer* 2021⁵) patients. Since these datasets contain biopsies taken from two time-points, they allow us to track clones in the same tumors over time (Fig. Q). We tested whether expression of the expansion signature pre-therapy could distinguish whether a clone expanded or contracted during PD-1 blockade. Consistent with our multi-site tumor mouse model data (in our response above), in all the datasets, clones that expanded over time had higher expression of the expansion signature pre-therapy (Fig. R). These results suggest that the expansion signature predicts clone expansion in external patient datasets too.

Q – Analysis of CD8⁺ T cell clones from longitudinal single-cell RNA/TCR-seq tumor biopsies from basal cell carcinoma (BCC) (Yost et al., 2019⁴), head and neck squamous cell carcinoma (HNSCC) (Luoma et al., 2022¹²), breast cancer (BC) (Bassez et al., 2021¹³) and non-small cell lung cancer (NSCLC) (Liu et al., 2021⁵) patient datasets pre (pre-therapy) and post (on-therapy) anti-PD-1 therapy. (Fig. 4c in paper)

R – Comparison of the mean expansion signature score of cells in the largest expanding and contracting clones pre-therapy. Analysis restricted to clones detected both pre and post therapy. (Fig. 4d in paper)

We thank the reviewer for this important point. We have added the results of this analysis, and corresponding descriptions in our manuscript on Fig 4c and 4d, and Page 10: Lines 1-6 as below.

We explored whether the expansion signature might be applicable to humans. We analyzed single-cell RNA/TCR-seq datasets obtained from site-matched longitudinal biopsies of basal cell carcinoma (BCC)⁴, head and neck squamous cell carcinoma (HNSCC)¹², breast cancer (BC)¹³ and non-small cell lung cancer (NSCLC)⁵ patient datasets pre and post anti-PD-1 therapy (Fig. 4c). In all the datasets, clones that expanded over time had higher expression of the expansion signature pre-therapy (Fig. 4d).

7-Could this expansion signature stratify TIL-ACT products based on expansion/clinical response post cell transfer? For this, the authors can use the dataset from Krishna et.al (DOI: 10.1126/science.abb9847) and also response to PD-1 blockade in Bassez, A. et al (<https://doi.org/10.1038/s41591-021-01323-8>)

We thank the reviewer for recognizing the potential of the expansion signature in stratifying patient responses and suggesting these additional analyses. The reviewer's suggestion to explore if the expansion signature can stratify TIL-ACT products based on the clinical response post cell transfer is of significant interest for the field, and we have therefore conducted the appropriate analysis. To enable thorough evaluation of the signature's power for stratifying responses, we chose to conduct our analysis on datasets that contained samples from numerous (at least 5 of each) responders and non-responders (a CTL019 therapy trial: Fraietta et al., *Nature Medicine* 2018¹⁴ and its commercial Tisa-cell therapy: Haradhvala et al., *Nature Medicine* 2022¹⁵). In both cases, we scored the transcriptomic data

of the pre-infusion CAR-T cell products for expression of the expansion gene signature. We then evaluated how this expansion score (pre-therapy) differed between responders and non-responders (Fig. S). Strikingly, we observed that pre-infusion products from responders had higher expansion scores than pre-infusion products from non-responders (Fig. Y-U). These results indicate that, in addition to monitoring responses to conventional immune checkpoint blockade therapy, the expansion signature may also have utility in stratifying responses to TIL-ACT therapy.

S – Pre-infusion CAR-T cell products were scored for their expression of the expansion signature. (Fig. 4h in paper)

T – Comparison of the mean expansion signature score of pre-infusion CAR-T cells between responding and non-responding patients undergoing CTL019 therapy (Fraietta et al., 2018¹⁴, n=31). (Fig. 4i in paper)

U – Comparison of the mean expansion signature score of pre-infusion CAR-T cells between responding and non-responding patients undergoing Tisa-cel therapy (Haradhvala et al., 2022¹⁵, n=13). (Extended Data Fig. 7h in paper)

We thank the reviewer for suggesting this additional analysis. We have added the results of this analysis, and corresponding descriptions in our manuscript on Fig 4h, 4i and Extended Data Fig. 7h. and Page 11: Lines 11-16 as below.

Lastly, we investigated whether the expansion signature had utility in predicting responses to CAR-T cell therapy. We scored pre-infusion CAR-T products for expression of the expansion signature and compared patient responses (Fig. 4h). In an early trial of tisagenlecleucel (CTL019)^{14,16}, preinfusion products expressed a significantly higher expansion score in responders than non-responders (Fig. 4i). A similar trend was confirmed in a separate dataset of tis-cel therapy¹⁵ (Extended Data Fig. 7h).

With regards to the reviewer’s suggestion of analyzing responses to PD-1 blockade in Bassez, A. et al *Nature Medicine* 2021¹³, please see our response to Point 7 – we have analyzed this dataset, and 3 other datasets of patients on PD-1 therapy^{4,5,12}, and shown the utility of the expansion signature in predicting clone expansion during PD-1 therapy.

8- From Fig.5 it seems very clear that the LAG3 is restringing the upregulation of the expansion signature. Therefore, I recommend the following experiment to formally demonstrate that this signature is associated with expansion.

Following the experimental set up of Fig.5a, I recommended to sort Lag3+ and Lag3- Pex cells, transfer them to new tumor-bearing host and treat both cohorts with/ PD-1 Blockade/control Ab and follow up expansion and tumor control, please see (<https://doi.org/10.1038/s41590-019-0312-6>)

We thank the reviewer for suggesting this additional experiment to formally demonstrate that the expansion signature is associated with expansion. We agree that the experiment suggested by the reviewer would provide additional data to support our findings. Unfortunately, however, the number of T_{pex} cells in our system limits our ability to perform the above experiment.

In the experiments of Miller et al., *Nature Immunology 2019*² (as cited by the reviewer), the researchers transfer 50,000 T_{pex} cells from mice 21 days after they have been inoculated with B16-OVA and subsequently vaccinated with GVAX-OVA cells (which combined, greatly increase the number of OVA-specific T_{pex} cells). They then assay the function of these cells *in vivo* against B16-OVA tumors. Following the experimental set up of Fig. 5a, we collect an average of 7942 Lag3+ T_{pex} cells (standard deviation of 2534 cells) per mouse, and much less (we estimate over 6-fold less) Lag3- T_{pex} cells on day 21. As a result, it is unfeasible for us to collect enough Lag3+ T_{pex}, let alone Lag3- T_{pex} cells to transfer into multiple new hosts under different conditions. Additionally, our model is designed to study the endogenous, polyclonal T cell response to tumors without the use of a defined antigen. While this approach more closely reflects the natural immune response to tumors, it also means that we would need to transfer far more cells to achieve the same level of tumor control as in antigen-specific systems. (i.e. B16-OVA system). Lastly, as in our response to Points 4 and 5, the overwhelming majority of LAG-3- precursor exhausted cells are not members of 'relevant' clones and are likely bystander cells. Consequently, it would be very difficult to size-match and directly compare the response of tumor-specific LAG-3+ and LAG-3- precursor exhausted cells.

However, given the reviewer's important point, we conducted additional *in vitro* experiments to directly assess if the expansion signature is associated with expansion. We focused on the differentiation states associated with 'relevant clones' – LAG-3+ precursor exhausted, intermediate exhausted and terminally exhausted differentiation states. In our original scRNA-seq dataset, we observed that the LAG-3+ precursor exhausted cells express the highest levels of expansion signature compared to the other exhausted cell states (Fig. V). We therefore sorted LAG-3+ precursor exhausted cells, intermediate and terminally exhausted cells, cultured them in IL-2 (100 U/mL) on plates coated with different concentrations of CD3 (1, 3.3 and 10 µg/mL) and CD28 (0.5 µg/mL). Cells were stained with CFSE, and their expansion was evaluated 72 hours after culture. In agreement with our observations that LAG-3+

precursor exhausted cells express the highest levels of the expansion signature, we found that LAG-3+ precursor exhausted cells also expand the most *in vitro* in all conditions, supporting the association between the expansion signature and expansion.

U – Expression of the expansion signature in the different T cell clusters

V – T cells from each cluster were sorted, stained with CFSE, and cultured in different concentrations of CD3. Cells were assayed at 72 hours.

Lastly, given the reviewer’s point, we have noted that due to experimental limitations, we do not demonstrate that the upregulation of the expansion signature is functionally associated with expansion, in our new **Limitations Section** on Page 15: Lines 17-20 as below.

In this study, we demonstrate a strong correlation between an ‘expansion’ gene signature and intratumoral clone expansion. Developing direct methods to prospectively identify clones with high expansion signatures would help determine whether upregulation of this signature functionally contributes to clonal expansion.

9- Do the expansion and tumor control is better upon combining LAG3- and TGFbeta blockade?

We thank the reviewer for raising this exciting point. In the dataset, we find that the TGF-β signaling signature is inversely correlated with the expansion signature, suggesting that it may play a role in limiting CD8+ T cell expansion. We agree with the reviewer that this finding has translational potential – and TGF-β blockade could be used in powerful combinational treatment with immune checkpoint

blockade therapies such as LAG-3. In this paper, our scope is on developing the multi-site tumor model, our finding of the expansion signature, and our demonstration of its predictive value for clone expansion. As such, we feel that further investigation of the potential uses of TGF- β blockade is out of scope. However, we agree with the reviewer that this is a promising path of investigation, and so to provide preliminary supportive data, we conducted an additional experiment testing the effect of TGF- β blockade on clonal T cell expansion in the tumor.

We administered a cohort of mice, inoculated with Lewis Lung carcinoma tumors, anti-TGF- β antibody (clone 1D11.16.8, Selleck) at 500 μ g/dose/mouse on days 14 and 16 and evaluated the number of PD-1⁺ CD8⁺ T cells found in the tumor on day 18 (Fig. X). Compared to untreated mice, blocking with anti-TGF- β suggested an increased trend in the number of T cells (Fig. Y,Z).

X – Mice, inoculated with Lewis Lung Carcinoma were treated with anti-TGF- β over 4 days.

Y – The number of PD-1⁺ CD8⁺ T cells in the tumor of mice resected 4 days (Left) and normalized per tumor mass (Right) after no treatment or anti-TGF- β antibody treatment.

Given the wide interest in the correlation between the expansion signature and TGF- β signaling, we thought it important to highlight the potential of these findings but also note the need for further experiments to functionally validate the association. We have added this to our new **Limitations Section** on Page 15: Lines 20-23 as below.

We also find several correlations between the expansion signature and gene signatures linked to biological signaling pathways and processes, such as TGF- β signaling. These pathways could be targeted to enhance intratumoral expansion, but further experiments are required to functionally validate these correlations.

Reviewer #2 (Remarks to the Author): with expertise in cancer immunology

In this study, Takahashi et al. perform a time-resolved analysis of clonal T cell dynamics in preclinical mouse tumor models. The experimental approach, combining a multi-site tumor model with mapping of clonal T cell responses by single-cell RNA-sequencing and TCR-sequencing, is elegant. Several of the findings on the dynamics of T cell expansion and contraction in tumor-bearing mice are interesting for our understanding of the dynamics of clonal T cell responses in general. However, what remains unclear is to which extent the authors T_{pex} expansion signature, identified in a non-treatment setting, can truly predict T cell expansion to cancer (pan-) immunotherapy. This main message of the paper remains to be thoroughly addressed to improve the manuscript and warrant its publication in Nature Communications.

We appreciate the reviewer's positive comments on our manuscript. We thank the reviewer for appreciating the experimental approach, and for highlighting that the findings on the dynamics of T cell expansion and contraction in tumor-bearing mice are interesting for our understanding of the dynamics of clonal T cell responses in general. We also appreciate the reviewer's coinage of our model as a 'multi-site tumor (mouse) model'. The reviewer's description better captures our model, and so we have accordingly altered all our references to our model in line with the reviewer's description.

The reviewer has raised several important points which we have clarified to strengthen our manuscript. Specifically, we have generated 2 additional scRNA/TCR-seq datasets and analyzed 6 additional patient datasets to support our message that the expansion signature can predict T cell expansion to cancer (pan-) immunotherapy, as below.

Major comments

1. In their LLC mouse cancer model, the authors identify a gene expression signature in T_{pex} that is (1) linked to the future expansion of T cell clones in absence of immune checkpoint blockade (Figure 3) and (2) upregulated in T cells expanding after anti-LAG-3 treatment in the same model (Figure 5). But can its analysis in T_{pex} before anti-LAG-3 treatment be used to predict clonal T cell expansion past anti-LAG3 treatment ? It is puzzling that the authors did not assess this, as the information should be contained in their datasets shown in Figure 5. This important point needs to be clarified to justify the authors claim that their signature is predictive of intratumoral T cell expansion across different immunotherapies, this important point needs to be a) clarified in the data from experiments with anti-LAG-3 treatment and b) extended to experiments with PD-1/CTLA4 blockade.

We thank the reviewer for raising this important point. We agree with the reviewer's suggestion that we should clarify whether the expansion signature is predictive of intratumoral T cell clone expansion across different immunotherapies. The reviewer suggests clarifying this claim with data from experiments with anti-LAG-3 treatment in Figure 5 and extending this to experiments with PD-1/CTLA4

blockade. Unfortunately, for the experiments in Figure 5, we either only collected bulk TCR-seq data, or bulk RNA-seq data on day 21 (the later timepoint), meaning we could not investigate whether the expansion signature is predictive of intratumoral T cell clone expansion. However, given the importance of the reviewer's point, we generated new datasets to corroborate our claims. Specifically, we generated an additional time-resolved scRNA/TCR-seq dataset using our multi-site tumor model whereby we collected transcriptomic/ bulk TCR-seq data on day 14, and bulk TCR-seq data on day 21, while the mice were treated with anti-LAG-3 (200 µg/dose/mouse) intraperitoneally on days 14 and 16. This new dataset enabled us to match the transcriptomic states of single CD8⁺ T cells before therapy, with their clonal expansion dynamics after anti-LAG-3 therapy. We similarly generated an additional time-resolved scRNA/TCR-seq dataset for anti-PD-1/CTLA-4 blockade (100 µg each/ dose/ mouse, dosed on days 14 and 16) blockade (Fig. A). We followed the same analysis pipeline as the original, untreated dataset to identify the relevant clusters in the dataset (Fig. B-E).

A – Tumors were resected from mice before and after anti-PDL1/CTLA4 and anti-LAG-3 treatment on days 14 and 21. Day 14 samples were processed by single cell RNA/TCR-seq and the remaining cells/ samples were processed through bulk TCR-seq. (Extended Data Fig. 6a in paper)

B – Uniform Manifold Approximation and Projection (UMAP) map of cells colored by clusters obtained from unsupervised Louvain clustering. (Extended Data Fig. 6b in paper)

C – Dot plot showing the expression of marker genes for each cluster. (Extended Data Fig. 6c in paper)

D – UMAP map of cells from tumors, colored and labelled by cell type. (Extended Data Fig. 6d in paper)

E – UMAP map of cells from tumors, colored and labelled by treatment. (Extended Data Fig. 6e in paper)

We next used this dataset to clarify whether the expansion signature is predictive of intratumoral T cell expansion across different immunotherapies. In line with our response to Reviewer 1 (Point 3), we tested whether the overall expression of the expansion signature in clones (approximated by weighting

each cell state's expansion signature score by the proportion of clonal cells in the state for each clone) could predict clone expansion (Fig. F). The overall expansion signature score before treatment strongly correlated with expansion during treatment under both anti-PDL1/CTLA-4 and anti-LAG-3 blockade (Fig. G). This correlation improved if we were more stringent with the clones we analyzed (Fig. H).

F – Clones were assigned a weighted average expansion signature score based on their gene expression across the three cell states at day 14. This score was then correlated with the degree of clonal expansion observed under untreated, anti-PD-L1/CTLA-4, and anti-LAG-3 treatment conditions. (Fig. 4a in paper)

G – Scatter plots comparing the overall expansion signature score with the expansion of clones in untreated (Left), anti-PDL1/CTLA4 (Middle) and anti-LAG-3 (Right) conditions. Analysis on clones with at least 3 cells in each cell state on day 14. (Fig. 4b in paper)

H – Results of correlations when analysis is restricted to clones with at least 5 cells in each cell state on day 14. (Extended Data Fig. 6g in paper)

We thank the reviewer for raising this important point – our new data supports the utility of the expansion signature in predicting intratumoral T cell expansion across different immunotherapies and significantly strengthens our manuscript. We have added the results of this analysis, and corresponding descriptions in our manuscript on Fig 4a-b and Extended Data Fig. 6, and Page 9: Lines 10-24 as below.

To establish the utility of the expansion signature, especially in the context of immunotherapy, we tested whether clonal expansion could be predicted from the expansion signature in both untreated and immunotherapy treated conditions (Fig. 4a). Using the multi-site tumor model, we generated additional time-resolved scRNA/TCR-seq datasets to track clonal responses under anti-PDL1/CTLA4 and anti-LAG-3 blockade. The new datasets enabled us to match the transcriptomic states of single CD8⁺ T cells before therapy, with their clonal expansion dynamics after therapy (Extended Data Fig. 6a). As before, unsupervised clustering of the single-cell RNA/TCR-seq data retrieved precursor exhausted, intermediate exhausted, terminally exhausted and proliferating cells (of which the majority were terminally exhausted cells) (Extended Data Fig. 6b-f). To approximate how clonal expression profiles would appear in non-enriched or human datasets, we calculated each clone's overall expansion signature score as the weighted average of its state-specific scores, using the proportion of clonal cells in each state on day 14 as weights. Strikingly, in untreated and in both anti-PDL-1/CTLA-4 and LAG-3 conditions, the clones' expansion signature score on day 14 correlated strongly with their future expansion (Fig. 4b), indicating that the expansion signature could be used to predict clone expansion during immunotherapy.

2. Related to the above: the authors do not find a pre-treatment enrichment of their expansion signature in human melanoma patients responding to anti-PD1/PD-L1 blockade (Extended data Figure 5d). This suggests that the signature, although as shown in Figure 4b, is induced by successful ICB treatment, does not allow to predict immunotherapy response in these cancer patients – These data argues for only limited power of the predictive signature in human cancer patients. They further are inconsistent with the authors findings for the basal cell carcinoma dataset shown in Figure 4a. Given these discrepancies, it seems important that the overall point of whether or not expression of the expansion signature in T_{pex} can or cannot predict immunotherapy remains unclear – and should be clarified by probing other scRNA-Seq datasets from human cancer patients to evaluate its predictive power.

We thank the reviewer for raising this important point. We agree with the reviewer that the manuscript would be significantly improved with additional analysis on human datasets to clarify whether the expansion signature can or cannot predict immunotherapy. As we describe in the introduction, the current limitations in sampling inherent to human biopsies make comprehensive analysis of clonal dynamics difficult. However, we have taken this opportunity to analyze 6 additional patient datasets (meaning we now have a total of 9 patient datasets in the manuscript).

First, we analyzed site-matched scRNA/TCR-seq biopsies taken from patients pre- and post-PD-1 blockade therapy. The datasets consisted of biopsies taken from basal cell carcinoma (BCC) (Yost et al., *Nature Medicine* 2019⁴), head and neck squamous cell carcinoma (HNSCC) (Luoma et al., *Cell* 2022¹²), breast cancer (BC) (Bassez et al., *Nature Medicine* 2021¹³) and non-small cell lung cancer (NSCLC) (Liu et al., *Nature Cancer* 2021⁵) patients. Since these datasets contain biopsies taken from two time-points,

they allow us to track clones in the same tumors over time (Fig. I). We tested whether expression of the expansion signature pre-therapy could distinguish whether a clone expanded or contracted during PD-1 blockade. Consistent with our multi-site tumor mouse model data (in our response above), in all the datasets, clones that expanded over time had higher expression of the expansion signature pre-therapy (Fig. J). These results suggest that the expansion signature can predict clone expansion during immunotherapy in patients as well as in mice.

F – Analysis of CD8⁺ T cell clones from longitudinal single-cell RNA/TCR-seq tumor biopsies from basal cell carcinoma (BCC) (Yost et al., 2019⁴), head and neck squamous cell carcinoma (HNSCC) (Luoma et al., 2022¹²), breast cancer (BC) (Bassez et al., 2021¹³) and non-small cell lung cancer (NSCLC) (Liu et al., 2021⁵) patient datasets pre (pre-therapy) and post (on-therapy) anti-PD-1 therapy. (Fig. 4c in paper)

J – Comparison of the mean expansion signature score of cells in the largest expanding and contracting clones pre-therapy. Analysis restricted to clones detected both pre and post therapy. (Fig. 4d in paper)

We next tested whether the expansion signature can predict patient response. When we compared responders and non-responders, clones from responders in post-treatment samples had significantly higher expansion signature scores than clones from non-responders (Fig. K) indicating that successful therapy was inducing clone expansion activity. Consistent with our previous results, no such difference was observed in pre-treatment samples where comparisons between responders and non-responders were possible (Fig. L). These results were consistent with our prior analysis on bulk RNA-seq melanoma tumor biopsies of pre- and on- PD-1/PD-L1 and PD-1/CTLA-4 blockade treatment (formerly Figure 4) where we showed that expression of the expansion signature in on-treatment samples could stratify responses.

K – Comparison of the mean expansion signature score of cells in clones from responding and non-responding patients on-therapy. (Fig. 4e in paper)

L – Comparison of the mean expansion signature score of cells in clones from responding and non-responding patients pre-therapy in datasets where comparisons could be made. (Extended Data Fig. 7a in paper)

We have added the results of this analysis, and corresponding descriptions in our manuscript on Fig 4c-e, and Extended Data Fig. 7a, and Page 10: Lines 1-12 as below.

We explored whether the expansion signature might be applicable to humans. We analyzed single-cell RNA/TCR-seq datasets obtained from site-matched longitudinal biopsies of basal cell carcinoma (BCC)⁴, head and neck squamous cell carcinoma (HNSCC)¹², breast cancer (BC)¹³ and non-small cell lung cancer (NSCLC)⁵ patient datasets pre and post anti-PD-1 therapy (Fig. 4c). In all the datasets, clones that expanded over time had higher expression of the expansion signature pre-therapy (Fig. 4d). Comparing responders and non-responders, clones from responders in post-treatment samples had significantly higher expansion signature scores than clones from non-responders (Fig. 4e). No such difference was observed in pre-treatment samples where comparisons between responders and non-responders were possible (Extended Data Fig. 7a).

Lastly, we explored if the expansion signature could stratify TIL-ACT products based on the clinical response post cell transfer. To enable thorough evaluation of the signature's power for stratifying responses, we chose to conduct our analysis on datasets that contained samples from responders and non-responders (a CTL019 therapy trial: Fraietta et al., *Nature Medicine* 2018¹⁴ and its commercial Tisa-cell therapy: Haradhvala et al., *Nature Medicine* 2022¹⁵). In both cases, we scored the transcriptomic data of the pre-infusion CAR-T cell products for expression of the expansion gene signature. We then evaluated how this expansion score (pre-therapy) differed between responders and non-responders (Fig. M). Strikingly, we observed that pre-infusion products from responders had higher expansion scores than pre-infusion products from non-responders (Fig. N-O). These results indicate that, in addition to monitoring responses to conventional immune checkpoint blockade therapy, the expansion signature also has utility in stratifying responses to TIL-ACT therapy.

M – Pre-infusion CAR-T cell products were scored for their expression of the expansion signature. (Fig. 4h in paper)

N – Comparison of the mean expansion signature score of pre-infusion CAR-T cells between responding and non-responding patients undergoing CTL019 therapy (Fraietta et al., 2018¹⁴, n=31). (Fig. 4i in paper)

O – Comparison of the mean expansion signature score of pre-infusion CAR-T cells between responding and non-responding patients undergoing Tisa-cel therapy (Haradhvala et al., 2022¹⁵, n=13). (Extended Data Fig. 7h in paper)

We have added the results of this analysis, and corresponding descriptions in our manuscript on Fig 4h-l, and Extended Data Fig. 7h. and Page 11: Lines 11-16 as below.

Lastly, we investigated whether the expansion signature had utility in predicting responses to CAR-T cell therapy. We scored pre-infusion CAR-T products for expression of the expansion signature and compared patient responses (Fig. 4h). In an early trial of tisagenlecleucel (CTL019)^{14,16}, preinfusion products expressed a significantly higher expansion score in responders than non-responders (Fig. 4i). A similar trend was confirmed in a separate dataset of tis-cel therapy¹⁵ (Extended Data Fig. 7h).

We thank the reviewer for raising this important point and providing us an opportunity to clarify the findings in our manuscript. Our data indicate that the expansion signature (1) in pre-therapy biopsies predicts clone expansion over PD-1 blockade, and (2) in on-therapy biopsies, stratifies patient responses to PD-1 blockade. With respect to clinical outcomes to immune checkpoint blockade therapy, these results suggest that clone expansion plays an important role in determining outcome after treatment has started. Other factors – such as the potential for recruitment of novel clones⁴, in addition to the potential of clone expansion in pre-existing clones, may play an important role in determining patient outcomes pre-therapy. Conversely, since we find that expression of the expansion signature in CAR-T cell infusion products (pre-treatment) corresponds with patient outcomes, in adoptive cell transfer therapy, the potential for clone expansion may play a more significant role in determining patient outcomes. We thought that this was an important to clarify for readers of our manuscript and so have added this to our **Discussions Section** in our manuscript on Page 14: Lines 9-17 as below.

In our study, we demonstrate that expression of the expansion signature in biopsies taken before immunotherapy treatment can predict clone expansion during treatment. We show that expression of the expansion signature in biopsies taken during immunotherapy treatment can stratify patient outcomes. With respect to clinical responses, these results suggest that clone expansion plays an important role in determining patient outcomes after treatment has started. Other factors – such as the potential for recruitment of novel clones⁴, in addition to the potential of clone expansion in pre-existing clones, may play an important role in predicting patient outcomes from biopsies taken before treatment. Interestingly, we find that expression of the expansion signature in CAR-T cell infusion products (pre-treatment) corresponds with patient outcomes.

3. The findings by the authors indicate that TGF-beta signaling may determine the capacity of Tpex cells to expand in tumors, which seems interesting with respect to the mechanisms limiting Tpex responses in tumors. Can the authors demonstrate in loss-of-function or gain-of-function experiments that TGF-beta impacts clonal T cell expansion in tumors?

We thank the reviewer for raising this interesting point. In the dataset, we find that the TGF- β signaling signature is inversely correlated with the expansion signature, suggesting that it may play a role in limiting CD8+ T cell expansion. We agree with the reviewer that the mechanisms limiting Tpex responses in tumors is highly interesting, and this would be the focus of our future studies. In this paper, our scope is on developing the multi-site tumor model, our finding of the expansion signature, and our demonstration of its predictive value for clone expansion. As such, we feel that further investigation of the mechanisms regulating clone contraction with TGF- β is out of scope. However, we agree with the reviewer that this is an interesting future path of investigation, and so to provide preliminary supportive data for these findings, we conducted an additional experiment testing the effect of loss-of-function TGF- β via antibody-blocking on clonal T cell expansion in the tumor.

We administered a cohort of mice, inoculated with Lewis Lung carcinoma tumors, anti-TGF- β antibody (clone 1D11.16.8, Selleck) at 500 $\mu\text{g}/\text{dose}/\text{mouse}$ on days 14 and 16 and evaluated the number of PD-1⁺ CD8⁺ T cells found in the tumor on day 18 (Fig. X). Compared to untreated mice, antibody blocking with anti-TGF- β suggested an increased trend in the number of T cells.

P – Mice, inoculated with Lewis Lung Carcinoma were treated with anti-TGF- β over 4 days.

Q – The number of PD-1⁺ CD8⁺ T cells in the tumor of mice resected 4 days (Left) and normalized per tumor mass (Right) after no treatment or anti-TGF- β antibody treatment.

Given the wide interest in the correlation between the expansion signature and TGF- β signaling, we thought it important to highlight the potential of these findings but also note the need for further experiments to functionally validate the association. We have added this to our new **Limitations Section** on Page 15: Lines 20-23 as below.

We also find several correlations between the expansion signature and gene signatures linked to biological signaling pathways and processes, such as TGF- β signaling. These pathways could be targeted to enhance intratumoral expansion, but further experiments are required to functionally validate these correlations.

Minor comments

4. Tsui et al., Nature 2022, have reported on CD62L⁺ T_{pex} driving clonal expansion. Is a signature for CD62L⁺ T_{pex} correlated with clonal T cell expansion in the authors dataset ?

We thank the reviewer for suggesting this additional analysis. We agree with the reviewer that analysis with the genesets derived in this paper would be of significant interest for the field. We have therefore assessed the correlation between the expansion signature and the signature derived from Myb⁺ T_{pex} cells, specifically the genesets derived from comparing CD62L⁺ with CD62L⁻ T_{pex} cells, and the genesets derived from comparing CD62L⁻ with Myb KO mice T_{pex} cells (Fig. R). We do not observe a consistent

correlation between the expansion signature score and the Myb+ Tpex cells. We observe a weak correlation between the expansion signature score and the CD62L- Tpex cell signature. The Myb+ Tpex signatures were generated from mice with chronic infections, where the effector site, and consequently the sampling site, is predominantly the spleen. This could make interpretations of gene signatures in the tumor difficult, particularly for transcriptomic signatures that may be affected by tissue location. For instance, in the tumor, we find that the majority of CD62L+ Tpex cells are from 'non-relevant' clones (clones with no TIM-3+ reads at either day 14 or 21) - projection of this cluster onto the ProjecTIL atlas⁶ suggests that these cells are likely naïve or bystander cells (cluster 4; Fig. S-V).

R – Scatter plots comparing the mean signature score of gene signatures from Tsui et al., 2022¹⁰ with the expansion of clones. Analysis on clones with at least 5 corresponding cells on day 14. (Extended Data Fig. 4f in paper)

S – UMAP map of cells colored by clusters obtained from unsupervised Louvain clustering.

T – Dot plot showing the expression of marker genes for each cluster – CD62L is enriched in cluster 4.

U – UMAP map of cells colored by whether they belong to ‘relevant clones’ (Clones with at least one read in the TIM-3 compartment at either day 14 or 21).

V – Projection of cluster 4 to the ProjecTILs reference atlas⁶

We have added the results from this analysis, and corresponding descriptions in our manuscript on Extended Data Fig 4f and Pages 8: Lines 10-12 as below.

Other previously reported precursor exhausted cell state gene signatures from mice and humans²⁹⁻³², including from a recently described CD62L+ precursor exhausted population³², did not consistently correlate with the expansion of clones (Extended Data Fig. 4e-f).

Reviewer #3 (Remarks to the Author): with expertise in computational immunology

Major comment

(1) If I understand correctly, when the authors did the experiments, the expanding clones and contracting clones were taken from the same 2 snapshots of times in the same mice, right? That means they are independent clones that occur simultaneously. But in Fig. 2e, the authors draw the plot in a way to indicate that T cell clones first expand and then contract in a sequential manner. There doesn't seem to be evidence to support this right? Why not contract first and then expand? As I mentioned, the contracting and expanding clones are different. The authors should conduct experiments at 3 instead of 2 time points, to define the expanding/contracting history of the same clone?

We thank the reviewer for raising this important point. We agree with the reviewer that for the experiments shown in Figure 2, the expanding clones and contracting clones were taken from the same 2 snapshots of times in the same mice. The plot in Fig. 2e indicate that T cell clones first expand and then contract in a sequential manner. We drew the T cell clonal dynamics in this way as our understanding of T cell immunity is that T cells first expand and then contract – with the understanding that the contracting clones must have first expanded (like the expanding clones we observe). However, we agree with the author that there is no evidence in the data in Figure 2 to suggest this to be the case. We have therefore corrected the figure to show that the expanding and contracting clones are different.

R – Characteristics of expanding and contracting clones. (Fig. 2e in paper)

(2) “These cells could not be differentiated based on their Uniform Manifold Approximation and Projection (UMAP) representation (Fig. 3c, Extended Data Fig. 3d). However, differential gene expression (DEG) analysis between precursor exhausted cells from expanding and contracting clones revealed that cells from expanding clones overexpressed 45 genes – of which 22 were canonical genes” UMAP cannot tell any difference, and there are only 22 genes that are differentially expressed. I worry the authors are reading too much into the data and the expansion signature is not very real.

We thank the reviewer for raising this important point. We understand the reviewer's concern with the derivation of the expansion signature and appreciate the reviewer for highlighting the need to test the signature in external datasets to see if it is 'real': if the expansion signature is not very real, we would

not expect it to correlate with expansion in external datasets. We have therefore generated two additional scRNA/TCR-seq datasets and conducted additional analysis on third-party human datasets to demonstrate that the expansion signature correlates with clone expansion in external datasets to address this concern.

First, we generated two additional time-resolved scRNA/TCR-seq dataset using our multi-site tumor model. We collected transcriptomic/ bulk TCR-seq data on day 14, and bulk TCR-seq data on day 21, while the mice were treated with anti-LAG-3 (200 $\mu\text{g}/\text{dose}/\text{mouse}$) or anti-PD-1/CTLA-4 (100 μg each/ dose/ mouse) blockade intraperitoneally on days 14 and 16. This new dataset enabled us to match the transcriptomic states of single CD8⁺ T cells before therapy, with their clonal expansion dynamics after anti-LAG-3 or anti-PD-1/CTLA-4 therapy. We used this dataset to evaluate whether the expansion signature correlates with clone expansion in separate murine datasets (this time, under immunotherapy). In line with our response to Reviewer 1 (Point 3), we tested whether the overall expression of the expansion signature in clones (approximated by weighting each cell state’s expansion signature score by the proportion of clonal cells in the state for each clone) could predict clone expansion (Fig. B). The overall expansion signature score before treatment strongly correlated with expansion during treatment in both the new datasets (under both anti-PDL1/CTLA-4 and anti-LAG-3 blockade) (Fig. C), demonstrating that the expansion signature correlates with expansion in external datasets.

B – Clones were assigned a weighted average expansion signature score based on their gene expression across the three cell states at day 14. This score was then correlated with the degree of clonal expansion observed under new datasets: anti-PD-L1/CTLA-4, and anti-LAG-3 treatment conditions. (Fig. 4a in paper)

C – Scatter plots comparing the overall expansion signature score with the expansion of clones in the original dataset (Left), new datasets: anti-PDL1/CTLA4 (Middle) and anti-LAG-3 (Right) treatment. Analysis on clones with at least 3 cells in each cell state on day 14. (Fig. 4b in paper)

We have added the results of this analysis, and corresponding descriptions in our manuscript on Fig 4a and 4b, and Page 9: Lines 10-24 as below.

To establish the utility of the expansion signature, especially in the context of immunotherapy, we tested whether clonal expansion could be predicted from the expansion signature in both untreated and immunotherapy treated conditions (Fig. 4a). Using the multi-site tumor model, we generated additional time-resolved scRNA/TCR-seq datasets to track clonal responses under anti-PDL1/CTLA4 and anti-LAG-3 blockade. The new datasets enabled us to match the transcriptomic states of single CD8⁺ T cells before therapy, with their clonal expansion dynamics after therapy (Extended Data Fig. 6a). As before, unsupervised clustering of the single-cell RNA/TCR-seq data retrieved precursor exhausted, intermediate exhausted, terminally exhausted and proliferating cells (of which the majority were terminally exhausted cells) (Extended Data Fig. 6b-f). To approximate how clonal expression profiles would appear in non-enriched or human datasets, we calculated each clone's overall expansion signature score as the weighted average of its state-specific scores, using the proportion of clonal cells in each state on day 14 as weights. Strikingly, in untreated and in both anti-PDL-1/CTLA-4 and LAG-3 conditions, the clones' expansion signature score on day 14 correlated strongly with their future expansion (Fig. 4b), indicating that the expansion signature could be used to predict clone expansion during immunotherapy.

Next, we tested our expansion signature on additional patient samples. We analyzed site-matched scRNA/TCR-seq biopsies taken from patients pre- and post-PD-1 blockade therapy. The datasets consisted of biopsies taken from basal cell carcinoma (BCC) (Yost et al., *Nature Medicine* 2019⁴), head and neck squamous cell carcinoma (HNSCC) (Luoma et al., *Cell* 2022¹²), breast cancer (BC) (Bassez et al., *Nature Medicine* 2021¹³) and non-small cell lung cancer (NSCLC) (Liu et al., *Nature Cancer* 2021⁵) patients. Since these datasets contain biopsies taken from two time-points, they allow us to track clones in the same tumors over time (Fig. D). We tested whether expression of the expansion signature pre-therapy could distinguish whether a clone expanded or contracted during PD-1 blockade. Consistent with our multi-site tumor mouse model data (in our response above), in all the datasets, clones that expanded over time had higher expression of the expansion signature pre-therapy (Fig. E). These results suggest that the expansion signature predicts clone expansion in external patient datasets too.

D – Analysis of CD8⁺ T cell clones from longitudinal single-cell RNA/TCR-seq tumor biopsies from basal cell carcinoma (BCC) (Yost et al., 2019⁴), head and neck squamous cell carcinoma (HNSCC) (Luoma et al., 2022¹²), breast cancer (BC) (Bassez et al., 2021¹³) and non-small cell lung cancer (NSCLC) (Liu et al., 2021⁵) patient datasets pre (pre-therapy) and post (on-therapy) anti-PD-1 therapy. (Fig. 4c in paper)

E – Comparison of the mean expansion signature score of cells in the largest expanding and contracting clones pre-therapy. Analysis restricted to clones detected both pre and post therapy. (Fig. 4d in paper)

We have added the results of this analysis, and corresponding descriptions in our manuscript on Fig 4c and 4d, and Page 10: Lines 1-6 as below.

We explored whether the expansion signature might be applicable to humans. We analyzed single-cell RNA/TCR-seq datasets obtained from site-matched longitudinal biopsies of basal cell carcinoma (BCC)⁴, head and neck squamous cell carcinoma (HNSCC)¹², breast cancer (BC)¹³ and non-small cell lung cancer (NSCLC)⁵ patient datasets pre and post anti-PD-1 therapy (Fig. 4c). In all the datasets, clones that expanded over time had higher expression of the expansion signature pre-therapy (Fig. 4d).

Lastly, in the analysis of our original derivation of the expansion signature, to reduce the likelihood that the signature is down to chance, we have added the results of a single cell weighted gene correlation network analysis (scWGCNA) on the precursor exhausted cells. Since scWGCNA clusters modules of gene networks based on co-expression similarity, we reasoned that this analysis would provide an alternative, unbiased assessment of the biological states. scWGCNA revealed 2 modules (Module 1 and 2). Module 1 contained genes associated with calcium ion release (Ryr2, Trdn) and DNA repair (Aptx, Ankhd1). Module 2 predominantly contained ribosomal genes, and gene ontology (GO) assessment associated these genes strongly with translation (Supplementary Table 2). Module 1 was highly expressed in cells from contracting clones whereas Module 2 was highly expressed in cells from expansion clones (Fig. F). Indeed, 24 of the 45 genes overexpressed in cells from expanding clones over contracting clones intersected with Module 2 genes (Fig. G). These results support our finding of a distinct gene signature upregulated in cells from expanding clones.

F – Precursor exhausted from expanding and contracting clones were scored for their expression of the Module 1 (Left) and 2 (Right) gene signatures obtained from single cell weighted gene co-expression network analysis (scWGCNA) analysis on day 14. (Extended Data Fig. 3f in paper)

G – Overlap in genes from Module 1 and genes overexpressed in cells from expanding clones. (Extended Data Fig. 4a in paper)

We have added the results of this analysis, and corresponding descriptions in our manuscript on Extended Data Fig 3f and Extended Data Fig. 4a, and Page 7: Lines 9-18 as below.

We first performed single cell weighted gene network correlation analysis (scWGCNA)^{17,18} – a method that clusters modules of gene networks based on co-expression similarity for an unbiased assessment of the biological states hidden within this precursor exhausted cluster. scWGCNA revealed 2 modules (Module 1 and 2). Module 1 contained genes associated with calcium ion release (Ryr2, Trdn) and DNA repair (Aptx, Ankd1). Module 2 predominantly contained ribosomal genes, and gene ontology (GO) assessment associated these genes strongly with translation (Supplementary Table 2). We scored cells from expanding and contracting clones for expression of genes from each module. Intriguingly, Module 1 was highly expressed in cells from contracting clones whereas Module 2 was highly expressed in cells from expansion clones (Extended Data Fig. 3e). These results suggested that T_{pex} cells from expanding and contracting clones were in distinct states. Differential gene expression (DEG) analysis between precursor exhausted cells from expanding and contracting clones revealed that cells from expanding clones overexpressed 45 genes of which 24 intersected with Module 2 genes (Extended Data Fig. 4a).

(3) “50 gene signatures of well-defined biological processes, and searched for correlations with the expansion 20 signature score at the clonal level.” If direct differential gene expression analyses only revealed 22 differentially expressed genes, how is it possible that the authors end up with so many hallmark pathways that are correlated? In other words, why were the genes in these hallmark pathways not found in the most direct differential gene expression analyses? This raises a concern of whether the observed correlations in this section of the result are simply a result of aggregation of random noises rather than true signal

We thank the reviewer for raising this important point. We understand the reviewer’s concern that the results of this section (Currently Extended Data Fig. 5) could simply be a result of aggregation of random noises rather than true signal. To address this concern, we first (1) re-ran the analysis removing any genes that overlapped between the expansion signature and the hallmark pathways (Fig. H), and (2) repeated this analysis to check for correlations in a separate dataset of antigen-signaled T cell clones in the tumor¹⁹ (Fig. I). We found similar correlations as we observed from our previous analysis. Hallmark gene pathways consistently correlated with the expansion signature score between the two datasets were: myc targets, oxidative phosphorylation, fatty acid metabolism and adipogenesis. These results

are consistent with previously reported associations between myc targets, oxidative phosphorylation, and fatty acid metabolism and T cell expansion²⁰⁻²², and as such, believe these results are not due to noise. We note that the differential gene expression was performed between expanding and contracting clones whereas this analysis was performed against the expansion score (to validate in different datasets) which may explain why these genes were not found in the differential gene expression analysis.

H Lewis Lung Carcinoma

I YUMMER1.7

H – Scatter plots comparing the mean expansion signature score of clones with various mean hallmark pathway signatures for our Lewis Lung Carcinoma dataset. (Extended Data Fig. 5a in paper)

I – Equivalent analysis on a YUMMER1.7 single cell dataset from Takahashi et al., *Science Immunology* 2024¹⁹ (Extended Data Fig. 5b in paper)

Hallmark pathway signatures which strongly correlated ($R^2 > 0.3$, $p < 0.05$) with the mean expansion signature score in the Lewis Lung Carcinoma dataset are shown. Analysis on clones with at least 5 precursor exhausted cells on day 14.

The hallmark pathway geneset analysis also revealed that the expansion score negatively correlates with TGF- β signaling in both datasets. We previously tested this correlation with other independent datasets of TGF- β signaling²³⁻²⁵ (current Extended Data Fig. 5c-d) and showed consistent results. Given the concern about these correlations being aggregations of noise, we also share a loss-of-function (by antibody blocking) validation experiment to investigate if TGF- β signaling is related to T cell expansion. We administered a cohort of mice, inoculated with Lewis Lung carcinoma tumors, anti-TGF- β antibody (clone 1D11.16.8, Selleck) at 500 μ g/dose/mouse on days 14 and 16 and evaluated the number of PD-1⁺ CD8⁺ T cells found in the tumor on day 18 (Fig. X). In support of our transcriptomic findings, blocking with anti-TGF- β suggested an increased trend in the number of T cells compared to untreated mice.

J – Mice, inoculated with Lewis Lung Carcinoma were treated with anti-TGF- β over 4 days.

K – The number of PD-1⁺ CD8⁺ T cells in the tumour of mice resected 4 days (Left), and normalized per tumor mass (Right) after no treatment or anti-TGF- β antibody treatment.

Given the reviewer’s concern with the correlation analysis, we thought it important to highlight the potential of these findings but also note the need for further experiments to functionally validate these associations. We have added this to our new **Limitations Section** on Page 15: Lines 20-23 as below.

We also find several correlations between the expansion signature and gene signatures linked to biological signaling pathways and processes, such as TGF- β signaling. These pathways could be targeted to enhance intratumoral expansion, but further experiments are required to functionally validate these correlations.

(4) “The expansion signature stratifies melanoma patient outcomes to PD-1 and PD-1/CTLA-4 9 blockade.” I feel the authors need to find at least one/ideally two other cohorts with similar data and repeat these analyses. See if the conclusions are robust

We thank the reviewer for making this important point. We have taken this opportunity to find 5 additional datasets from immunotherapy patients to see if the expansion signature stratifies patient

outcomes. First, we analyzed scRNA/TCR-seq biopsies taken from patients on PD-1 blockade therapy. The datasets, which we also utilized to see if our expansion signature predicted clone expansion in response to Point 1, consisted of biopsies taken from basal cell carcinoma (BCC) (Yost et al., *Nature Medicine* 2019⁴), head and neck squamous cell carcinoma (HNSCC) (Luoma et al., *Cell* 2022¹²), breast cancer (BC) (Bassez et al., *Nature Medicine* 2021¹³) and non-small cell lung cancer (NSCLC) (Liu et al., *Nature Cancer* 2021⁵) patients. When we compared responders and non-responders, for three out of the four datasets, clones from responders in on-therapy samples had significantly higher expansion signature scores than clones from non-responders (Fig. L) indicating that successful therapy was inducing clone expansion activity. These results were consistent with our prior analysis on bulk RNA-seq melanoma tumor biopsies of pre- and on- PD-1/PD-L1 and PD-1/CTLA-4 blockade treatment (formerly Figure 4) where we showed that expression of the expansion signature in on-treatment samples could stratify responses.

L – Comparison of the mean expansion signature score of cells in clones from responding and non-responding patients on-therapy. (Fig. 4e in paper)

We have added the results of this analysis, and corresponding descriptions in our manuscript on Fig 4e. and Page 10: Lines 6-8 as below.

Comparing responders and non-responders, clones from responders in post-treatment samples had significantly higher expansion signature scores than clones from non-responders (Fig. 4e).

Next, we explored if the expansion signature could stratify TIL-ACT products based on the clinical response post cell transfer. We analyzed datasets that contained samples from responders and non-responders (a CTL019 therapy trial: Fraietta et al., *Nature Medicine* 2018¹⁴ and its commercial Tisa-cell therapy: Haradhvala et al., *Nature Medicine* 2022¹⁵). In both cases, we scored the transcriptomic data of the pre-infusion CAR-T cell products for expression of the expansion gene signature. We then evaluated how this expansion score (pre-therapy) differed between responders and non-responders

(Fig. M). Interestingly, we observed that pre-infusion products from responders had higher expansion scores than pre-infusion products from non-responders (Fig. N-O).

M – Pre-infusion CAR-T cell products were scored for their expression of the expansion signature. (Fig. 4h in paper)

N – Comparison of the mean expansion signature score of pre-infusion CAR-T cells between responding and non-responding patients undergoing CTL019 therapy (Fraietta et al., 2018¹⁴, n=31). (Fig. 4i in paper)

O – Comparison of the mean expansion signature score of pre-infusion CAR-T cells between responding and non-responding patients undergoing Tisa-cel therapy (Haradhvala et al., 2022¹⁵, n=13). (Extended Data Fig. 7h in paper)

We have added the results of this analysis, and corresponding descriptions in our manuscript on Fig 4h-I and Extended Data Fig. 7h, and Page 11: Lines 11-16 as below.

Lastly, we investigated whether the expansion signature had utility in predicting responses to CAR-T cell therapy. We scored pre-infusion CAR-T products for expression of the expansion signature and compared patient responses (Fig. 4h). In an early trial of tisagenlecleucel (CTL019)^{14,16}, preinfusion products expressed a significantly higher expansion score in responders than non-responders (Fig. 4i). A similar trend was confirmed in a separate dataset of tis-cel therapy¹⁵ (Extended Data Fig. 7h).

(5) “Together, these results demonstrate that LAG-3 blockade 14 enhances the expansion signature and re-expands contracted clones in the tumor.” I might have missed this. But I don’t seem to get why LAG-3 blockade would preferentially re-expand contracting clones as opposed to expanding clones. Could the authors explain, in the context of their theory?

We thank the reviewer for raising this point. In our system, we observe that LAG-3 blockade increases the fraction and total frequency of expanding clones (Fig. 5e, Extended Data Fig. 9c). Most of these clones are clones that expanded between days 14 and 21 (Fig. 5f). Compared to untreated mice, we observe a significant difference in the fraction of clones that contracted between days 14 and 21, that are expanding between days 21 and 28 (Fig. 5g). Similarly, we observe that this contributes to the

increase in the total frequency of clones expanding during LAG-3 blockade (Extended Data Fig. 9d). While there is a similar, but non-significant trend in the fraction and frequency of expanded clones that expand between days 21 and 28 (Fig. 5f, Extended Data Fig. 9c), we agree with the reviewer that it is interesting that LAG-3 blockade might be preferentially re-expanding contracting clones as opposed to expanding clones. We postulate that this might relate to the potential effects of TGF- β on clonal dynamics, and how LAG-3 blockade reduces this signaling (Fig. 5c). Clonal contraction is associated with increased TGF- β signaling (Extended Data Fig. 5). In contracting clones, LAG-3 may be reinforcing TGF- β mediated quiescence or limiting TCR signaling that would normally counteract TGF- β effects. With LAG-3 blockade, these contracting clones would be relieved of one major brake, tipping the balance away from contraction (TGF- β) towards re-expansion. Expanding clones have lower TGF- β signaling and so their clonal dynamics may be less affected by LAG-3 blockade.

We think these are interesting points to highlight for discussion and have therefore added these ideas to our **Discussions Section** in our manuscript on Page 14: Lines 1-7 as below.

We observe the greatest effect of LAG-3 blockade on re-expanding contracting clones. TGF- β signaling is highest in contracting clones and its suppressive effect, may in part be reinforced by LAG-3. With LAG-3 blockade, the contracting clones could be preferentially relieved of one major brake, tipping their dynamics from contraction towards re-expansion. Future mechanistic studies using the multi-site tumor model could explore how immune checkpoints differentially regulate the expansion and contraction phases, guiding the development of strategies to selectively enhance phase-specific immune responses.

Minor comments

(1) “the sizes of any given clone (as identified by the read 19 frequency of a TCR sequence in the repertoire) were equivalent” I agree the TCR frequencies are highly correlated. But “equivalent” is a big exaggeration.

We thank the reviewer for highlighting this miswording. We agree with the reviewer that the TCR frequencies are highly correlated. We have corrected the sentence by replacing ‘equivalent’ with ‘highly correlated’, as below.

*Across all tumors, including the left and right flank tumors, the sizes of any given clone (as identified by the read frequency of a TCR sequence in the repertoire) were **highly correlated** (Fig. 1a-b, Extended Data Fig. 1c).*

(2) “We observed a consistent trend in the fraction of large clones 3 that were contracting (Extended Data Fig. 7b).” “Consistent trend” seems somewhat confusing. Will “similar fraction” or something like

this be more accurate and straightforward? Is this what the authors mean?

We thank the reviewer for highlighting this confusing phrase. This is indeed what we meant. We have corrected the sentence by replacing 'consistent trend' with 'similar difference', as we wanted to convey that the differences in the fractions are trending in the same manner.

We observed a similar difference in the fraction of large clones that were contracting (Extended Data Fig. 9b).

- 1 Zehn, D., Thimme, R., Lugli, E., de Almeida, G. P. & Oxenius, A. 'Stem-like' precursors are the fount to sustain persistent CD8(+) T cell responses. *Nat Immunol* **23**, 836-847 (2022). <https://doi.org/10.1038/s41590-022-01219-w>
- 2 Miller, B. C. *et al.* Subsets of exhausted CD8(+) T cells differentially mediate tumor control and respond to checkpoint blockade. *Nat Immunol* **20**, 326-336 (2019). <https://doi.org/10.1038/s41590-019-0312-6>
- 3 Im, S. J. *et al.* Characteristics and anatomic location of PD-1(+)TCF1(+) stem-like CD8 T cells in chronic viral infection and cancer. *Proc Natl Acad Sci U S A* **120**, e2221985120 (2023). <https://doi.org/10.1073/pnas.2221985120>
- 4 Yost, K. E. *et al.* Clonal replacement of tumor-specific T cells following PD-1 blockade. *Nat Med* **25**, 1251-1259 (2019). <https://doi.org/10.1038/s41591-019-0522-3>
- 5 Liu, B. *et al.* Temporal single-cell tracing reveals clonal revival and expansion of precursor exhausted T cells during anti-PD-1 therapy in lung cancer. *Nat Cancer* **3**, 108-121 (2022). <https://doi.org/10.1038/s43018-021-00292-8>
- 6 Andreatta, M. *et al.* Interpretation of T cell states from single-cell transcriptomics data using reference atlases. *Nat Commun* **12**, 2965 (2021). <https://doi.org/10.1038/s41467-021-23324-4>
- 7 Kataoka, H. *et al.* FTY720, sphingosine 1-phosphate receptor modulator, ameliorates experimental autoimmune encephalomyelitis by inhibition of T cell infiltration. *Cell Mol Immunol* **2**, 439-448 (2005).
- 8 Im, S. J. *et al.* Defining CD8+ T cells that provide the proliferative burst after PD-1 therapy. *Nature* **537**, 417-421 (2016). <https://doi.org/10.1038/nature19330>
- 9 Utzschneider, D. T. *et al.* Early precursor T cells establish and propagate T cell exhaustion in chronic infection. *Nat Immunol* **21**, 1256-1266 (2020). <https://doi.org/10.1038/s41590-020-0760-z>
- 10 Tsui, C. *et al.* MYB orchestrates T cell exhaustion and response to checkpoint inhibition. *Nature* **609**, 354-360 (2022). <https://doi.org/10.1038/s41586-022-05105-1>
- 11 Kallies, A., Zehn, D. & Utzschneider, D. T. Precursor exhausted T cells: key to successful immunotherapy? *Nat Rev Immunol* **20**, 128-136 (2020). <https://doi.org/10.1038/s41577-019-0223-z>

- 12 Luoma, A. M. *et al.* Tissue-resident memory and circulating T cells are early responders to pre-surgical cancer immunotherapy. *Cell* **185**, 2918-2935.e2929 (2022). <https://doi.org/10.1016/j.cell.2022.06.018>
- 13 Bassez, A. *et al.* A single-cell map of intratumoral changes during anti-PD1 treatment of patients with breast cancer. *Nat Med* **27**, 820-832 (2021). <https://doi.org/10.1038/s41591-021-01323-8>
- 14 Fraietta, J. A. *et al.* Determinants of response and resistance to CD19 chimeric antigen receptor (CAR) T cell therapy of chronic lymphocytic leukemia. *Nat Med* **24**, 563-571 (2018). <https://doi.org/10.1038/s41591-018-0010-1>
- 15 Haradhvala, N. J. *et al.* Distinct cellular dynamics associated with response to CAR-T therapy for refractory B cell lymphoma. *Nat Med* **28**, 1848-1859 (2022). <https://doi.org/10.1038/s41591-022-01959-0>
- 16 Doan, A. E. *et al.* FOXO1 is a master regulator of memory programming in CAR T cells. *Nature* **629**, 211-218 (2024). <https://doi.org/10.1038/s41586-024-07300-8>
- 17 Morabito, S., Reese, F., Rahimzadeh, N., Miyoshi, E. & Swarup, V. hdWGCNA identifies co-expression networks in high-dimensional transcriptomics data. *Cell Rep Methods* **3**, 100498 (2023). <https://doi.org/10.1016/j.crmeth.2023.100498>
- 18 Langfelder, P. & Horvath, S. WGCNA: an R package for weighted correlation network analysis. *BMC Bioinformatics* **9**, 559 (2008). <https://doi.org/10.1186/1471-2105-9-559>
- 19 Takahashi, M. *et al.* Intratumoral antigen signaling traps CD8(+) T cells to confine exhaustion to the tumor site. *Sci Immunol* **9**, eade2094 (2024). <https://doi.org/10.1126/sciimmunol.ade2094>
- 20 Vardhana, S. A. *et al.* Impaired mitochondrial oxidative phosphorylation limits the self-renewal of T cells exposed to persistent antigen. *Nat Immunol* **21**, 1022-1033 (2020). <https://doi.org/10.1038/s41590-020-0725-2>
- 21 Zhang, Y. *et al.* Enhancing CD8(+) T Cell Fatty Acid Catabolism within a Metabolically Challenging Tumor Microenvironment Increases the Efficacy of Melanoma Immunotherapy. *Cancer Cell* **32**, 377-391.e379 (2017). <https://doi.org/10.1016/j.ccell.2017.08.004>
- 22 Araki, K. *et al.* Translation is actively regulated during the differentiation of CD8(+) effector T cells. *Nat Immunol* **18**, 1046-1057 (2017). <https://doi.org/10.1038/ni.3795>
- 23 Gabriel, S. S. *et al.* Transforming growth factor- β -regulated mTOR activity preserves cellular metabolism to maintain long-term T cell responses in chronic infection. *Immunity* **54**, 1698-1714.e1695 (2021). <https://doi.org/10.1016/j.immuni.2021.06.007>
- 24 Wells, A. C. *et al.* Let-7 enhances murine anti-tumor CD8 T cell responses by promoting memory and antagonizing terminal differentiation. *Nat Commun* **14**, 5585 (2023). <https://doi.org/10.1038/s41467-023-40959-7>
- 25 Nath, A. P. *et al.* Comparative analysis reveals a role for TGF- β in shaping the residency-related transcriptional signature in tissue-resident memory CD8+ T cells. *PLoS One* **14**, e0210495 (2019). <https://doi.org/10.1371/journal.pone.0210495>

REVIEWER COMMENTS

Reviewer #1 (Remarks to the Author):

The reviewer express satisfaction with the revision. The authors have effectively addressed the majority of my primary concerns. I have a few minor suggestions for their consideration:

We thank the reviewer for their positive comments on our manuscript. We have now included the additional figures as suggested by the reviewer as below.

a) Relative to my Q1, In addition to the projection, I would recommend to calculate a pseudo-time along the T_{pex}-Tex differentiation trajectory, to better illustrate the potential transitional nature of the PD-1+TIM-3+ Ly108+ CD8+ TILs

We thank the reviewer for this suggestion. We have calculated the relevant pseudo-time and included this in the manuscript as Extended Data Fig. 3e.

b) Clarify whether the expansion signature is more enriched in proliferating vs. non-proliferating Tex_terms (I did not see this clarification in the text of as part of the answer to Q3)

We thank the reviewer for this suggestion. We have clarified whether the expansion signature is more enriched in proliferating vs non-proliferating Tex-terms and included this analysis in the manuscript as Extended Data Fig. 4g.

Reviewer #2 (Remarks to the Author):

The authors have succesfully adressed all my concerns raised at first review. The new data in the revised paper significantly improve the manuscript which to me now is acceptable for publication. I congratulate the authors on a very interesting and thorough study.

Reviewer #3 (Remarks to the Author):

The authors have addressed all my concerns successfully. Thank you!

[name redacted]

We sincerely thank all reviewers for their positive feedback and encouraging words. We are pleased that the additional data strengthened the manuscript and that it is now deemed suitable for publication.